# A Global Convergence Theory for Deep ReLU Implicit Networks via Over-parameterization

**Tianxiang Gao**
Department of Computer Science
Iowa State University
gaotx@iastate.edu

**Hailiang Liu**
Department of Mathematics
Iowa State University
hliu@iastate.edu

**Jia Liu**
Dept. of Electrical and Computer Engineering
The Ohio State University
liu@ece.osu.edu

**Hridesh Rajan**
Department of Computer Science
Iowa State University
hridesh@iastate.edu

**Hongyang Gao**
Department of Computer Science
Iowa State University
hygao@iastate.edu

## Abstract

Implicit deep learning has received increasing attention recently, since it generalizes the recursive prediction rules of many commonly used neural network architectures. Its prediction rule is provided implicitly based on the solution of an equilibrium equation. Although many recent studies have experimentally demonstrates its superior performances, the theoretical understanding of implicit neural networks is limited. In general, the equilibrium equation may not be well-posed during the training. As a result, there is no guarantee that a vanilla (stochastic) gradient descent (SGD) training nonlinear implicit neural networks can converge. This paper fills the gap by analyzing the gradient flow of Rectified Linear Unit (ReLU) activated implicit neural networks. For an $m$-width implicit neural network with ReLU activation and $n$ training samples, we show that a randomly initialized gradient descent converges to a global minimum at a linear rate for the square loss function if the implicit neural network is *over-parameterized*. It is worth noting that, unlike existing works on the convergence of (S)GD on finite-layer over-parameterized neural networks, our convergence results hold for implicit neural networks, where the number of layers is *infinite*.

## 1 Introduction

**1) Background and Motivation:** In the last decade, implicit deep learning (El Ghaoui et al., 2019) have attracted more and more attention. Its popularity is mainly because it generalizes the recursive rules of many widely used neural network architectures. A line of recent works (Bai et al., 2019; El Ghaoui et al., 2019; Bai et al., 2020) have shown that the implicit neural network architecture is a wider class that includes most current neural network architectures as special cases, such as feed-forward neural networks, convolution neural networks, residual networks, and recurrent neural networks. Moreover, implicit deep learning is also well known for its competitive performance compared to other regular deep neural networks but using significantly fewer computational resources (Dabre & Fujita, 2019; Dehghani et al., 2018; Bai et al., 2018).

Although a line of literature has been shown the superior performance of implicit neural networks experimentally, the theoretical understanding is still limited. To date, it is still unknown if a simple first-order optimization method such as (stochastic) gradient descent can converge on an implicit neural network activated by a nonlinear function. Unlike a regular deep neural network, an implicit neural network could have infinitely many layers, resulting in the possibility of divergence of the

forward propagation (El Ghaoui et al., 2019; Kawaguchi, 2021). The main challenge in establishing the convergence of implicit neural network training lies in the fact that, in general, the equilibrium equation of implicit neural networks cannot be solved in closed-form. What exacerbates the problem is the well-posedness of the forward propagation. In other words, the equilibrium equation may have zero or multiple solutions. A line of recent studies have suggested a number of strategies to handle this well-posedness challenge, but they all involved reformulation or solving subproblems in each iteration. For example, El Ghaoui et al. (2019) suggested to reformulate the training as the Fenchel divergence formulation and solve the reformulated optimization problem by projected gradient descent method. However, this requires solving a projection subproblem in each iteration and convergence was only demonstrated numerically. By using an extra softmax layer, Kawaguchi (2021) established global convergence result of gradient descent for a linear implicit neural network. Unfortunately, their result cannot be extended to nonlinear activations, which are critical to the learnability of deep neural networks.

This paper proposes a global convergence theory of gradient descent for implicit neural networks activated by nonlinear Rectified Linear Unit (ReLU) activation function by using overparameterization. Specifically, we show that random initialized gradient descent with fixed stepsize converges to a global minimum of a ReLU implicit neural network at a linear rate as long as the implicit neural network is *overparameterized*. Recently, over-parameterization has been shown to be effective in optimizing finite-depth neural networks (Zou et al., 2020; Nguyen & Mondelli, 2020; Arora et al., 2019; Oymak & Soltanolkotabi, 2020). Although the objective function in the training is non-smooth and non-convex, it can be shown that GD or SGD converge to a global minimum linearly if the width $m$ of each layer is a polynomial of the number of training sample $n$ and the number of layers $h$, *i.e.*, $m = \text{poly}(n, h)$. However, these results cannot be directly applied to implicit neural networks, since implicit neural networks have infinitely many hidden layers, *i.e.*, $h \to \infty$, and the well-posedness problem surfaces during the training process. In fact, Chen et al. (2018); Bai et al. (2019; 2020) have all observed that the time and number of iterations spent on forward propagation are gradually increased with the the training epochs. Thus, we have to ensure the unique equilibrium point always exists throughout the training given that the width $m$ is only polynomial of $n$.

**2) Preliminaries of Implicit Deep Learning:** In this work, we consider an implicit neural network with the transition at the $\ell$-th layer in the following form (El Ghaoui et al., 2019; Bai et al., 2019):

$$z^\ell = \sigma\left(\frac{\gamma}{\sqrt{m}}A z^{\ell-1} + \phi(x)\right), \tag{1}$$

where $\phi : \mathbb{R}^d \to \mathbb{R}^m$ is a feature mapping function that transforms an input vector $x \in \mathbb{R}^d$ to a desired feature vector $\phi \triangleq \phi(x)$, $z^\ell \in \mathbb{R}^m$ is the output of the $\ell$-th layer, $A \in \mathbb{R}^{m \times m}$ is a trainable weight matrix, $\sigma(u) = \max\{0, u\}$ is the ReLU activation function, and $\gamma \in (0, 1)$ is a fixed scalar to scale $A$. As will be shown later in Section 2.1, $\gamma$ plays the role of ensuring the existence of the limit $z^* = \lim_{\ell \to \infty} z^\ell$. In general, the feature mapping function $\phi$ is a nonlinear function, which extracts features from the low-dimensional input vector $x$. In this paper, we consider a simple nonlinear feature mapping function $\phi$ given by

$$\phi(x) \triangleq \frac{1}{\sqrt{m}}\sigma(Wx), \tag{2}$$

where $W \in \mathbb{R}^{m \times d}$ is a trainable parameter matrix. As $\ell \to \infty$, an implicit neural network can be considered an *infinitely* deep neural network. Consequently, $z^*$ is not only the limit of the sequence $\{z^\ell\}_{\ell=0}^\infty$ with $z_0 = 0$, but it is also the *equilibrium point* (or *fixed point*) of the equilibrium equation:

$$z^* = \sigma(\tilde{\gamma} A z^* + \phi), \tag{3}$$

where $\tilde{\gamma} \triangleq \gamma/\sqrt{m}$. In implicit neural networks, the prediction $\hat{y}$ for the input vector $x$ is the combination of the fixed point $z^*$ and the feature vector $\phi$, *i.e.*,

$$\hat{y} = u^T z^* + v^T \phi, \tag{4}$$

where $u, v \in \mathbb{R}^m$ are trainable weight vectors. For simplicity, we use $\theta \triangleq \text{vec}(A, W, u, v)$ to group all training parameters. Given a training data set $\{(x_i, y_i)\}_{i=1}^n$, we want to minimize

$$L(\theta) = \sum_{i=1}^n \frac{1}{2}(\hat{y}_i - y_i)^2 = \frac{1}{2}\|\hat{y} - y\|^2, \tag{5}$$

where $\hat{y}$ and $y$ are the vectors formed by stacking all the prediction and labels.

**3) Main Results:** Our results are based on the following observations. We first analyze the forward propagation and find that the unique equilibrium point always exists if the scaled matrix $\tilde{\gamma}\boldsymbol{A}$ in Eq. (3) has an operator norm less than one. Thus, the well-posedness problem is reduced to finding a sequence of scalars $\{\gamma_k\}_{k=1}^{\infty}$ such that $\tilde{\gamma}_k\boldsymbol{A}(k)$ is appropriately scaled. To achieve this goal, we show that the operator norm $\boldsymbol{A}(k)$ is uniformly upper bounded by a constant over all iterations. Consequently, a fixed scalar $\gamma$ is enough to ensure the well-posedness of Eq. (3). Our second observation is from the analysis of the gradient descent method with infinitesimal step-size (gradient flow). By applying the chain rule with the gradient flow, we derive the dynamics of prediction $\hat{\boldsymbol{y}}(t)$ which is governed by the spectral property of a Gram matrix. In particular, if the smallest eigenvalue of the Gram matrix is lower bounded throughout the training, the gradient descent method enjoys a linear convergence rate. Along with some basic functional analysis results, it can be shown that the smallest eigenvalue of the Gram matrix at initialization is lower bounded if no two data samples are parallel. Although the Gram matrix varies in each iteration, the spectral property is preserved if the Gram matrix is close to its initialization. Thus, the convergence problem is reduced to showing the Gram matrix in latter iterations is close to its initialization. Our third observation is that we find random initialization, over-parameterization, and linear convergence jointly enforce the (operator) norms of parameters upper bounded by some constants and close to their initialization. Accordingly, we can use this property to show that the operator norm of $\boldsymbol{A}$ is upper bounded and the spectral property of the Gram matrix is preserved throughout the training. Combining all these insights together, we can conclude that the random initialized gradient descent method with a constant step-size converges to a global minimum of the implicit neural network with ReLU activation.

The main contributions of this paper are summarized as follows:

(i) By scaling the weight matrix $\boldsymbol{A}$ with a fixed scalar $\gamma$, we show that the unique equilibrium point $\boldsymbol{z}^*$ for each $\boldsymbol{x}$ always exists during the training if the parameters are randomly initialized, even for the nonlinear ReLU activation function.

(ii) We analyze the gradient flow of implicit neural networks. Despite the non-smooth and non-convexity of the objective function, the convergence to a global minimum at a linear rate is guaranteed if the implicit neural network is over-parameterized and the data is non-degenerate.

(iii) Since gradient descent is discretized version of gradient flow, we can show gradient descent with fixed stepsize converges to a global minimum of implicit neural networks at a linear rate under the same assumptions made by the gradient flow analysis, as long as the stepsize is chosen small enough.

**Notation**: For a vector $\boldsymbol{x}$, $\|\boldsymbol{x}\|$ is the Euclidean norm of $\boldsymbol{x}$. For a matrix $\boldsymbol{A}$, $\|\boldsymbol{A}\|$ is the operator norm of $\boldsymbol{A}$. If $\boldsymbol{A}$ is a square matrix, then $\lambda_{\min}(\boldsymbol{A})$ and $\lambda_{\max}(\boldsymbol{A})$ denote the smallest and largest eigenvalue of $\boldsymbol{A}$, respectively, and $\lambda_{\max}(\boldsymbol{A}) \leq \|\boldsymbol{A}\|$. We denote $[n] \triangleq \{1, 2, \cdots, n\}$.

## 2 WELL-POSEDNESS AND GRADIENT COMPUTATION

In this section, we provide a simple condition for the equilibrium equation (3) to be well-posed in the sense that the unique equilibrium point exists. Instead of backpropagating through all the intermediate iterations of a forward pass, we derive the gradients of trainable parameters by using the implicit function theorem. In this work, we make the following assumption on parameter initialization.

**Assumption 1** (Random Initialization). The entries $\boldsymbol{A}_{ij}$ and $\boldsymbol{W}_{ij}$ are randomly initialized by the standard Gaussian distribution $\mathcal{N}(0, 1)$, and $\boldsymbol{u}_i$ and $\boldsymbol{v}_i$ are randomly initialized by the symmetric Bernoulli or Rademacher distribution.

**Remark 2.1.** This initialization is similar to the approaches widely used in practice (Glorot & Bengio, 2010; He et al., 2015). The result obtained in this work can be easily extended to the case where the distributions for $\boldsymbol{A}_{ij}$, $\boldsymbol{W}_{ij}$, $\boldsymbol{u}_i$, and $\boldsymbol{v}_i$ are replaced by *sub-Gaussian random variables*.

### 2.1 FORWARD PROPAGATION AND WELL-POSEDNESS

In a general implicit neural network, Eq. (3) is not necessarily well-posed, since it may admit zero or multiple solutions. In this work, we show that scaling the matrix $\boldsymbol{A}$ with $\tilde{\gamma} = \gamma/\sqrt{m}$ guarantees

the existence and uniqueness of the equilibrium point $z^*$ with random initialization. This follows from a foundational result in random matrix theory as restated in the following lemma.

**Lemma 2.1** (Vershynin (2018), Theorem 4.4.5)**.** Let $A$ be an $m \times n$ random matrix whose entries $A_{ij}$ are independent, zero-mean, and sub-Gaussian random variables. Then, for any $t > 0$, we have $\|A\| \leq CK(\sqrt{m} + \sqrt{n} + t)$ with probability at least $1 - 2e^{-t^2}$. Here $C > 0$ is a fixed constant, and $K = \max_{i,j} \|A_{ij}\|_{\psi_2}$.

Under Assumption 1, Lemma 2.1 implies that, with exponentially high probability, $\|A\| \leq c\sqrt{m}$ for some constant $c > 0$. By scaling $A$ by a positive scalar $\tilde{\gamma}$, we show that the transition Eq. (1) is a contraction mapping. Thus, the unique equilibrium point exists with detailed proof in Appendix A.1.

**Lemma 2.2.** If $\|A\| \leq c\sqrt{m}$ for some $c > 0$, then for any $\gamma_0 \in (0, 1)$, the scalar $\gamma \triangleq \min\{\gamma_0, \gamma_0/c\}$ uniquely determines the existence of the equilibrium $z^*$ for every $x$, and $\|z^\ell\| \leq \frac{1}{1-\gamma_0}\|\phi\|$ for all $\ell$.

Lemma 2.2 indicates that equilibria always exist if we can maintain the operator norm of the scaled matrix $(\gamma/\sqrt{m})A$ less than 1 during the training. However, the operator norms of matrix $A$ are changed by the update of the gradient descent. It is hard to use a fixed scalar $\gamma$ to scale the matrix $A$ over all iterations, unless the operator norm of $A$ is bounded. In Section 3, we will show $\|A\| \leq 2c\sqrt{m}$ always holds throughout the training, provided that $\|A(0)\| \leq c\sqrt{m}$ at initialization and the width $m$ is sufficiently large. Thus, by using the scalar $\gamma = \min\{\gamma_0, \gamma_0/(2c)\}$ for any $\gamma_0 \in (0, 1)$, equilibria always exist and the equilibrium equation Eq. (3) is well-posed.

## 2.2 BACKWARD GRADIENT COMPUTING

For a regular network with finite layers, one needs to store all intermediate parameters and apply backpropagation to compute the gradients. In contrast, for an implicit neural network, we can derive the formula of the gradients by using the implicit function theorem. Here, the equilibrium point $z^*$ is a root of the function $f$ given by $f(z, A, W) \triangleq z - \sigma(\tilde{\gamma}Az + \phi)$. The essential challenge is to ensure the partial derivative $\partial f/\partial z$ at $z^*$ is always invertible throughout the training. The following lemma shows that the partial derivative $\partial f/\partial z$ at $z^*$ always exists with the scalar $\gamma$, and the gradient derived in the lemma always exists. The gradient derivations for each trainable parameters are provided in the following lemma and the proof is provided in Appendix A.2.

**Lemma 2.3** (Gradients of an Implicit Neural Network)**.** Define the scalar $\gamma \triangleq \min\{\gamma_0, \gamma_0/c\}$. If $\|A\| \leq c\sqrt{m}$ for some constant $c > 0$, then for any $\gamma_0 \in (0, 1)$, the following results hold:

(i) The partial derivatives of $f$ with respect to $z$, $A$, and $W$ are

$$\frac{\partial f}{\partial z} = [I_m - \tilde{\gamma}\, \mathbf{diag}(\sigma'(\tilde{\gamma}Az + \phi))A]^T, \tag{6}$$

$$\frac{\partial f}{\partial A} = -\tilde{\gamma}\left[z^T \otimes \mathbf{diag}(\sigma'(\tilde{\gamma}Az + \phi))\right]^T, \tag{7}$$

$$\frac{\partial f}{\partial W} = -\frac{1}{\sqrt{m}}\left[x^T \otimes \mathbf{diag}(\sigma'(Wx))\right]^T \mathbf{diag}(\sigma'(\tilde{\gamma}Az + \phi))^T, \tag{8}$$

where $\tilde{\gamma} \triangleq \gamma/\sqrt{m}$.

(ii) For any vector $v$, the following inequality holds

$$\lambda_{\min}\left\{I_m - \tilde{\gamma}\, \mathbf{diag}(\sigma'(v))A\right\} > 1 - \gamma_0 > 0, \tag{9}$$

which further implies $\partial f/\partial z$ is invertible at the equilibrium point $z^*$.

(iii) The gradient of the objective function $L$ with respect to $A$ and $W$ are given by

$$\nabla_u L = \sum_{i=1}^n (\hat{y}_i - y_i)z_i, \qquad \nabla_v L = \sum_{i=1}^n (\hat{y}_i - y_i)\phi_i, \tag{10}$$

$$\nabla_A L = \sum_{i=1}^n \tilde{\gamma}(\hat{y}_i - y_i)D_i^T\left[I_m - \tilde{\gamma}D_i A\right]^{-T} u z_i^T, \tag{11}$$

$$\nabla_W L = \sum_{i=1}^n \frac{1}{\sqrt{m}}(\hat{y}_i - y_i)E_i^T\left\{D_i^T\left[I_m - \tilde{\gamma}D_i A\right]^{-T} u + v\right\}x_i^T, \tag{12}$$

where $\boldsymbol{z}_i$ is the equilibrium point for the training data $(\boldsymbol{x}_i, y_i)$, $\boldsymbol{D}_i \triangleq \mathbf{diag}(\sigma'(\tilde{\gamma}\boldsymbol{A}\boldsymbol{z}_i + \boldsymbol{\phi}_i))$, and $\boldsymbol{E}_i \triangleq \mathbf{diag}(\sigma'(\boldsymbol{W}\boldsymbol{x}_i))$.

# 3 GLOBAL CONVERGENCE OF THE GRADIENT DESCENT METHOD

In this section, we establish the global convergence results of the gradient descent method in implicit neural networks. In Section 3.1, we first study the dynamics of the prediction induced by the gradient flow, that is, the gradient descent with infinitesimal step-size. We show that the dynamics of the prediction is controlled by a Gram matrix whose smallest eigenvalue is strictly positive over iterations with high probability. Based on the findings in gradient flow, we will show that the random initialized gradient descent method with a constant step-size converges to a global minimum at a linear rate in Section 3.2.

## 3.1 CONTINUOUS TIME ANALYSIS

The gradient flow is equivalent to the gradient descent method with an infinitesimal step-size. Thus, the analysis of gradient flow can serve as a stepping stone towards understanding discrete-time gradient-based algorithms. Following previous works on gradient flow of different machine learning models (Saxe et al., 2013; Du et al., 2019; Arora et al., 2018; Kawaguchi, 2021) the gradient flow of implicit neural networks is given by the following ordinary differential equations:

$$\frac{d\text{vec}\,(\boldsymbol{A})}{dt} = -\frac{\partial L(t)}{\partial \boldsymbol{A}}, \quad \frac{d\text{vec}\,(\boldsymbol{W})}{dt} = -\frac{\partial L(t)}{\partial \boldsymbol{W}}, \quad \frac{d\boldsymbol{u}}{dt} = -\frac{\partial L(t)}{\partial \boldsymbol{u}}, \quad \frac{d\boldsymbol{v}}{dt} = -\frac{\partial L(t)}{\partial \boldsymbol{v}}, \quad (13)$$

where $L(t)$ represents the value of the objective function $L(\boldsymbol{\theta})$ at time $t$.

Our results relies on the analysis of the dynamics of prediction $\hat{\boldsymbol{y}}(t)$. In particular, Lemma 3.1 shows that the dynamics of prediction $\hat{\boldsymbol{y}}(t)$ is governed by the spectral property of a Gram matrix $\boldsymbol{H}(t)$.

**Lemma 3.1** (Dynamics of Prediction $\hat{\boldsymbol{y}}(t)$). Suppose $\|\boldsymbol{A}(s)\| \leq c\sqrt{m}$ for all $0 \leq s \leq t$. Let $\boldsymbol{X} \in \mathbb{R}^{n \times d}$, $\boldsymbol{\Phi}(s) \in \mathbb{R}^{n \times m}$, and $\boldsymbol{Z}(s) \in \mathbb{R}^{n \times m}$ be the matrices whose rows are the training data $\boldsymbol{x}_i$, feature vectors $\boldsymbol{\phi}_i$, and equilibrium points $\boldsymbol{z}_i$ at time $s$, respectively. With scalar $\gamma \triangleq \min\{\gamma_0, \gamma_0/c\}$ for any $\gamma_0 \in (0, 1)$, the dynamics of prediction $\hat{\boldsymbol{y}}(t)$ is given by

$$\frac{d\hat{\boldsymbol{y}}}{dt} = - \left[(\gamma^2 \boldsymbol{M}(t) + \boldsymbol{I}_n) \circ \boldsymbol{Z}(t)\boldsymbol{Z}(t)^T + \boldsymbol{Q}(t) \circ \boldsymbol{X}\boldsymbol{X}^T + \boldsymbol{\Phi}(t)\boldsymbol{\Phi}(t)^T\right] (\hat{\boldsymbol{y}} - \boldsymbol{y}),$$
$$\triangleq - \boldsymbol{H}(t)(\hat{\boldsymbol{y}} - \boldsymbol{y}), \quad (14)$$

where $\circ$ is the Hadamard product (i.e., element-wise product), and matrices $\boldsymbol{M}(t) \in \mathbb{R}^{n \times n}$ and $\boldsymbol{Q}(t) \in \mathbb{R}^{n \times n}$ are defined as follows:

$$\boldsymbol{M}(t)_{ij} \triangleq \frac{1}{m}\boldsymbol{u}^T(I_m - \tilde{\gamma}\boldsymbol{D}_i\boldsymbol{A})^{-1}\boldsymbol{D}_i\boldsymbol{D}_j^T(I_m - \tilde{\gamma}\boldsymbol{D}_j\boldsymbol{A})^{-T}\boldsymbol{u}, \quad (15)$$

$$\boldsymbol{Q}(t)_{ij} \triangleq \frac{1}{m} \left(\boldsymbol{D}_i^T(I_m - \tilde{\gamma}\boldsymbol{D}_i\boldsymbol{A})^{-1}\boldsymbol{u} + \boldsymbol{v}\right)^T \boldsymbol{E}_i\boldsymbol{E}_j^T \left(\boldsymbol{D}_j^T(I_m - \tilde{\gamma}\boldsymbol{D}_j\boldsymbol{A})^{-T}\boldsymbol{u} + \boldsymbol{v}\right). \quad (16)$$

Note that the matrix $\boldsymbol{H}(t)$ is clearly positive semidefinite since it is the sum of three positive semidefinite matrices. If there exists a constant $\lambda > 0$ such that $\lambda_{\min}\{\boldsymbol{H}(t)\} \geq \lambda > 0$ for all $t$, *i.e.*, $\boldsymbol{H}(t)$ is positive definite, then the dynamics of the loss function $L(t)$ satisfies the following inequality

$$L(t) \leq \exp\{-\lambda t\}L(0),$$

which immediately indicates that the objective value $L(t)$ is consistently decreasing to zero at a geometric rate. With random initialization, we will show that $\boldsymbol{H}(t)$ is positive definite as long as the number of parameters $m$ is sufficiently large and no two data points are parallel to each other. In particular, by using the nonlinearity of the feature map $\phi$, we will show that the smallest eigenvalue of the Gram matrix $\boldsymbol{G}(t) \triangleq \boldsymbol{\Phi}(t)\boldsymbol{\Phi}(t)^T$ is strictly positive over all time $t$ with high probability. As a result, the smallest eigenvalue of $\boldsymbol{H}(t)$ is always strictly positive.

Clearly, $\boldsymbol{G}(t)$ is a time-varying matrix. We first analyze its spectral property at its initialization. When $t = 0$, it follows from the definition of the feature vector $\phi$ in Eq. (2) that

$$\boldsymbol{G}(0) = \boldsymbol{\Phi}(0)\boldsymbol{\Phi}(0)^T = \frac{1}{m}\sigma(\boldsymbol{X}\boldsymbol{W}(0)^T)\sigma(\boldsymbol{X}\boldsymbol{W}(0)^T)^T = \frac{1}{m}\sum_{r=1}^{m}\sigma(\boldsymbol{X}\boldsymbol{w}_r(0))\sigma(\boldsymbol{X}\boldsymbol{w}_r(0))^T.$$

By Assumption 1, each vector $\boldsymbol{w}_r(0)$ follows the standard multivariate normal distribution, *i.e.*, $\boldsymbol{w}_r(0) \sim N(\boldsymbol{0}, \boldsymbol{I}_d)$. By letting $m \to \infty$, we obtain the covariance matrix $\boldsymbol{G}^\infty \in \mathbb{R}^{n \times n}$ as follows:

$$\boldsymbol{G}^\infty \triangleq \mathbb{E}_{\boldsymbol{w} \sim N(\boldsymbol{0}, \boldsymbol{I}_d)}[\sigma(\boldsymbol{X}\boldsymbol{w})\sigma(\boldsymbol{X}\boldsymbol{w})^T]. \tag{17}$$

Here $\boldsymbol{G}^\infty$ is a Gram matrix induced by the ReLU activation function and the random initialization. The following lemma shows that the smallest eigenvalue of $\boldsymbol{G}^\infty$ is strictly positive as long as no two data points are parallel. Moreover, later in Lemmas 3.3 and 3.4, we conclude that the spectral property of $\boldsymbol{G}^\infty$ is preserved in the Gram matrices $\boldsymbol{G}(0)$ and $\boldsymbol{G}(t)$ during the training, as long as the number of parameter $m$ is sufficiently large.

**Lemma 3.2.** Assume $\|\boldsymbol{x}_i\| = 1$ for all $i \in [n]$. If $\boldsymbol{x}_i \nparallel \boldsymbol{x}_j$ for all $i \neq j$, then $\lambda_0 \triangleq \lambda_{\min}\{\boldsymbol{G}^\infty\} > 0$.

*Proof.* The proof follows from the *Hermite Expansions* of the matrix $\boldsymbol{G}^\infty$ and the complete proof is provided in Appendix A.4. $\square$

**Assumption 2** (Training Data). Without loss of generality, we can assume that each $\boldsymbol{x}_i$ is normalized to have a unit norm, *i.e.*, $\|\boldsymbol{x}_i\| = 1$, for all $i \in [n]$. Moreover, we assume $\boldsymbol{x}_i \nparallel \boldsymbol{x}_j$ for all $i \neq j$.

In most real-world datasets, it is extremely rare that two data samples are parallel. If this happens, by adding some random noise perturbation to the data samples, Assumption 2 can still be satisfied. Next, we show that at the initialization, the spectral property of $\boldsymbol{G}^\infty$ is preserved in $\boldsymbol{G}(0)$ if $m$ is sufficiently large. Specifically, the following lemma shows that if $m = \tilde{\Omega}(n^2)$, then $\boldsymbol{G}(0)$ has a strictly positive smallest eigenvalue with high probability. The proof follows from the standard concentration bound for Gaussian random variables, and we relegate the proof to Appendix A.5.

**Lemma 3.3.** Let $\lambda_0 = \lambda_{\min}(\boldsymbol{G}^\infty) > 0$. If $m = \Omega\left(\frac{n^2}{\lambda_0^2} \log\left(\frac{n}{\delta}\right)\right)$, then with probability of at least $1 - \delta$, it holds that $\|\boldsymbol{G}(0) - \boldsymbol{G}^\infty\|_2 \leq \frac{\lambda_0}{4}$, and hence $\lambda_{\min}(\boldsymbol{G}(0)) \geq \frac{3}{4}\lambda_0$.

During training, $\boldsymbol{G}(t)$ is time-varying, but it can be shown to be close to $\boldsymbol{G}(0)$ and preserve the spectral property of $\boldsymbol{G}^\infty$, if the matrices $\boldsymbol{W}(t)$ has a bounded operator norm and it is not far away from $\boldsymbol{W}(0)$. This result is formally stated below and its proof is provided in Appendix A.6.

**Lemma 3.4.** Suppose $\|\boldsymbol{W}(0)\| \leq c\sqrt{m}$, and $\lambda_{\min}\{\boldsymbol{G}(0)\} \geq \frac{3}{4}\lambda_0$. For any matrix $\boldsymbol{W} \in \mathbb{R}^{m \times d}$ that satisfies $\|\boldsymbol{W}\| \leq 2c\sqrt{m}$ and $\|\boldsymbol{W} - \boldsymbol{W}(0)\| \leq \frac{\sqrt{m}\lambda_0}{16c\|\boldsymbol{X}\|^2} \triangleq R$, the matrix defined by $\boldsymbol{G} \triangleq \frac{1}{m}\sigma(\boldsymbol{X}\boldsymbol{W}^T)\sigma(\boldsymbol{X}\boldsymbol{W}^T)^T$ satisfies $\|\boldsymbol{G} - \boldsymbol{G}(0)\| \leq \frac{\lambda_0}{4}$ and $\lambda_{\min}(\boldsymbol{G}) \geq \frac{\lambda_0}{2}$.

The next lemma shows three facts: (1) The smallest eigenvalue of $\boldsymbol{G}(t)$ is strictly positive for all $t \geq 0$; (2) The objective value $L(t)$ converges to zero at a linear convergence rate; (3) The (operator) norms of all trainable parameters are upper bounded by some constants, which further implies that unique equilibrium points in matrices $\boldsymbol{Z}(t)$ always exist. We prove this lemma by induction, which is provided in Appendix A.7.

**Lemma 3.5.** Suppose that $\|\boldsymbol{u}(0)\| = \sqrt{m}$, $\|\boldsymbol{v}(0)\| = \sqrt{m}$, $\|\boldsymbol{W}(0)\| \leq c\sqrt{m}$, $\|\boldsymbol{A}(0)\| \leq c\sqrt{m}$, and $\lambda_{\min}\{\boldsymbol{G}(0)\} \geq \frac{3}{4}\lambda_0 > 0$. If $m = \Omega\left(\frac{c^2 n \|\boldsymbol{X}\|^2}{\lambda_0^3}\|\hat{\boldsymbol{y}}(0) - \boldsymbol{y}\|^2\right)$ and $0 < \gamma \leq \min\{\frac{1}{2}, \frac{1}{4c}\}$, then for any $t \geq 0$, the following results hold:

   (i) $\lambda_{\min}(\boldsymbol{G}(t)) \geq \frac{\lambda_0}{2}$,

  (ii) $\|\boldsymbol{u}(t)\| \leq \frac{16c\sqrt{n}}{\lambda_0}\|\hat{\boldsymbol{y}}(0) - \boldsymbol{y}\|$,

 (iii) $\|\boldsymbol{v}(t)\| \leq \frac{8c\sqrt{n}}{\lambda_0}\|\hat{\boldsymbol{y}}(0) - \boldsymbol{y}\|$,

 (iv) $\|\boldsymbol{W}(t)\| \leq 2c\sqrt{m}$,

  (v) $\|\boldsymbol{A}(t)\| \leq 2c\sqrt{m}$,

 (vi) $\|\hat{\boldsymbol{y}}(t) - \boldsymbol{y}\|^2 \leq \exp\{-\lambda_0 t\}\|\hat{\boldsymbol{y}}(0) - \boldsymbol{y}\|^2$.

By using simple union bounds in Lemma 2.1 and 3.5, we immediately obtain the global convergence result for the gradient flow as follows.

**Theorem 3.1** (Convergence Rate of Gradient Flow). Suppose that Assumptions 1 and 2 hold. If we set the number of parameter $m = \Omega\left(\frac{n^2}{\lambda_0^2}\log\left(\frac{n}{\delta}\right)\right)$ and choose $0 < \gamma \leq \min\{\frac{1}{2}, \frac{1}{4c}\}$, then with probability at least $1 - \delta$ over the initialization, we have

$$\|\hat{\boldsymbol{y}}(t) - \boldsymbol{y}\|^2 \leq \exp\{-\lambda_0 t\}\|\hat{\boldsymbol{y}}(0) - \boldsymbol{y}\|^2, \quad \forall t \geq 0.$$

This theorem establishes the global convergence of the gradient flow. Despite the nonconvexity of the objective function $L(\boldsymbol{\theta})$, Theorem 3.1 shows that if $m$ is sufficiently large, then the objective value is consistently decreasing to zero at a geometric rate. In particular, Theorem 3.1 requires $m = \tilde{\Omega}(n^2)$, which is similar or even better than recent results for the neural network with *finite* layers (Nguyen & Mondelli, 2020; Oymak & Soltanolkotabi, 2020; Allen-Zhu et al., 2019; Zou & Gu, 2019; Du et al., 2019). In particular, previous results showed that $m$ is a polynomial of the number of training sample $n$ and the number of layers $h$, *i.e.*, $m = \Omega(n^\alpha h^\beta)$ with $\alpha \geq 2$ and $\beta \geq 12$ (Nguyen & Mondelli, 2020, Table 1). These results do not apply in our case since we have *infinitely* many layers. By taking advantage of the nonlinear feature mapping function, we establish the global convergence for the gradient flow with $m$ independent of depth $h$.

## 3.2 DISCRETE TIME ANALYSIS

In this section, we show that the randomly initialized gradient descent method with a fixed step-size converges to a global minimum at a linear rate. With similar argument used in the analysis of gradient flow, we can show the (operator) norms of the training parameters are upper bounded by some constants. It is worth noting that, unlike the analysis of gradient flow, we do not have an explicit formula for the dynamics of prediction $\hat{\boldsymbol{y}}(t)$ in the discrete time analysis. Instead, we have to show the difference between the equilibrium points $\boldsymbol{Z}(k)$ in two consecutive iterations are bounded. Based on this, we can further bound the changes in the predictions $\hat{y}(k)$ between two consecutive iterations. Another challenge is to show the objective value consistently decreases over iterations. A general strategy is to show the objective function is (semi-)smooth with respect to parameter $\boldsymbol{\theta}$, and apply the descent lemma to the objective function (Nguyen & Mondelli, 2020; Allen-Zhu et al., 2019; Zou & Gu, 2019). In this section, we take advantage of the nonlinear feature mapping function in Eq. (4). Consequently, we are able to obtain a Polyak-Łojasiewicz-like condition (Karimi et al., 2016; Nguyen, 2021), which allows us to provide a much simpler proof.

The following lemma establishes the global convergence for the gradient descent method with a fixed step-size when the operator norms of $\boldsymbol{A}(0)$ and $\boldsymbol{W}(0)$ are bounded and $\lambda_{\min}(\boldsymbol{G}(0)) > 0$. The proof is proved in Appendix A.8.

**Lemma 3.6** (Gradient Descent Convergence Rate). Suppose $\|\boldsymbol{u}(0)\| = \sqrt{m}$, $\|\boldsymbol{v}(0)\| = \sqrt{m}$, $\|\boldsymbol{W}(0)\| \leq c\sqrt{m}$, $\|\boldsymbol{A}(0)\| \leq c\sqrt{m}$, and $\lambda_{\min}\{\boldsymbol{G}(0)\} \geq \frac{3}{4}\lambda_0 > 0$. If $0 < \gamma \leq \min\{\frac{1}{2}, \frac{1}{4c}\}$, $m = \Omega\left(\frac{c^2 n\|\boldsymbol{X}\|^2}{\lambda_0^3}\|\hat{\boldsymbol{y}}(0) - \boldsymbol{y}\|^2\right)$, and stepsize $\alpha = \mathcal{O}\left(\lambda_0/n^2\right)$, then for any $k \geq 0$, we have

   (i) $\lambda_{\min}(\boldsymbol{G}(k)) \geq \frac{\lambda_0}{2}$,

   (ii) $\|\boldsymbol{u}(k)\| \leq \frac{32c\sqrt{n}}{\lambda_0}\|\hat{\boldsymbol{y}}(0) - \boldsymbol{y}\|$,

   (iii) $\|\boldsymbol{v}(k)\| \leq \frac{16c\sqrt{n}}{\lambda_0}\|\hat{\boldsymbol{y}}(0) - \boldsymbol{y}\|$,

   (iv) $\|\boldsymbol{W}(k)\| \leq 2c\sqrt{m}$,

   (v) $\|\boldsymbol{A}(k)\| \leq 2c\sqrt{m}$,

   (vi) $\|\hat{\boldsymbol{y}}(k) - \boldsymbol{y}\|^2 \leq (1 - \alpha\lambda_0/2)^k\|\hat{\boldsymbol{y}}(0) - \boldsymbol{y}\|^2$.

By using simple union bounds to combine Lemma 2.1 and 3.6, we obtain the global convergence result for the gradient descent.

**Theorem 3.2** (Convergence Rate of Gradient Descent). Suppose that Assumption 1 and 2 hold. If we set $m = \Omega\left(\frac{n^2}{\lambda_0}\log\left(\frac{n}{\delta}\right)\right)$, $0 < \gamma \leq \min\{\frac{1}{2}, \frac{1}{4c}\}$, and choose step-size $\alpha = \mathcal{O}\left(\lambda_0/n^2\right)$, then with probability at least $1 - \delta$ over the initialization, we have

$$\|\hat{\boldsymbol{y}}(k) - \boldsymbol{y}\|^2 \leq (1 - \alpha\lambda_0/2)^k\|\hat{\boldsymbol{y}}(0) - \boldsymbol{y}\|^2, \quad \forall k \geq 0.$$

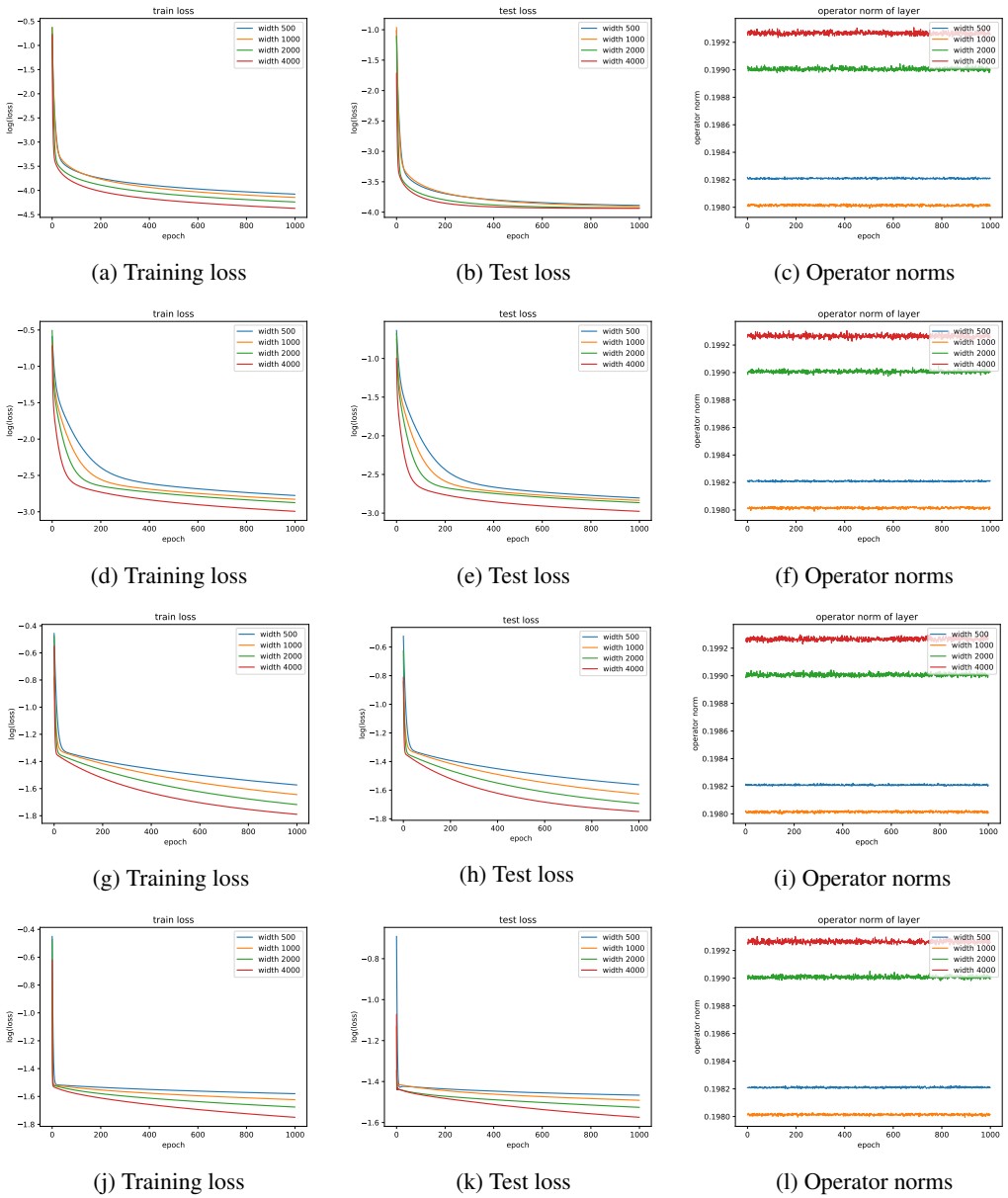

Figure 1: Results on MNIST, FashionMNIST, CIFAR10, and SVHN. We evaluate the impact of the width $m$ on the training loss, test loss, and operator norm of the scaled matrix $(\gamma/\sqrt{m})\boldsymbol{A}(k)$ on four real datasets.

## 4 EXPERIMENTAL RESULTS

In this section, we use real-world datasets MNST, FashionMNST, CIFAR10, and SVHN to evaluate our theoretical findings. We initialize the entries of parameters $\boldsymbol{A}$, $\boldsymbol{W}$, $\boldsymbol{u}$, and $\boldsymbol{v}$ by standard Gaussian or symmetric Bernoulli distribution independently as suggested in Assumption 1. For each dataset, we only use classes 0 and 1, and 500 samples are randomly drawn from each class to generate the training dataset with $n = 1000$ samples. All data samples are converted to gray scale and resized into a $28 \times 28$ pixel image. We also normalize each data to have unit norm. If two parallel samples are observed, we add a random Gaussian noise perturbation to one of them. Thus, Assumption 2 is also satisfied. We run 1000 epochs of gradient descent with a fixed step-size. We test three metrics with different width $m$. We first test how the extent of over-parameterization

affects the convergence rates. Then, we test the relation between the extent of over-parameterization and the operator norms between matrix $\boldsymbol{A}(k)$ and its initialization. Note that the "operator norm" in the plots denotes the operator norm of the scaled matrix $(\gamma/\sqrt{m})\|\boldsymbol{A}(k)\|$. Third, we test the extent of over-parameterization and the performance of the trained neural network on the unseen test data. Similar to the training dataset, We randomly select 500 samples from each class as the test dataset.

From Figure 1, the figures in the first column show that as $m$ becomes larger, we have better convergence rates. The figures in the second column show that as $m$ becomes larger, the neural networks achieve lower test loss. The figures in the third column show that the operator norms are slightly larger for larger $m$ but overall the operator norms are approximately equal to its initialization. The bell curve from the classical bias-variance trade-off does not appear. This opens the door to a new research direction in implicit neural networks in the analyses of generalization error and bias-variance trade-off.

## 5 RELATED WORKS

Implicit models has been explored explored by the deep learning community for decades. For example, Pineda (1987) and ALMEIDA (1987) studied implicit differentiation techniques for training recurrent dynamics, also known as *recurrent back-propagation* (Liao et al., 2018). Recently, there has been renewed interested in the implicit models in the deep learning community (El Ghaoui et al., 2019; Gould et al., 2019). For example, Bai et al. (2019) introduces an implicit neural network called *deep equilibrium model* for the for the task of sequence modeling. By using implicit ODE solvers, Chen et al. (2018) proposed *neural ordinary differential equation* as an implicit residual network with continuous-depth. Other instantiations of implicit modeling include optimization layers (Djolonga & Krause, 2017; Amos & Kolter, 2017), differentiable physics engines (de Avila Belbute-Peres et al., 2018; Qiao et al., 2020), logical structure learning (Wang et al., 2019), and continuous generative models (Grathwohl et al., 2019).

The theoretical study of training finite-layer neural networks via over-parameterization has been an active research area. Jacot et al. (2018) showed the trajectory of the gradient descent method can be characterized by a kernel called *neural tangent kernel* for smooth activation and infinitely width neural networks. For a finite-width neural network with smooth or ReLU activation, Arora et al. (2019); Du et al. (2019); Li & Liang (2018) showed that the dynamics of the neural network is governed by a Gram matrix. Consequently, Zou et al. (2020); Du et al. (2019); Allen-Zhu et al. (2019); Nguyen & Mondelli (2020); Arora et al. (2019); Zou & Gu (2019); Oymak & Soltanolkotabi (2020) showed that (stochastic) gradient descent can attain global convergence for training a finite-layer neural network when the width $m$ is a polynomial of the sample size $n$ and the depth $h$. However, their results cannot be applied directly to implicit neural networks since implicit neural networks have infinite layers and the equilibrium equation may not be well-posed. Our work establishes the well-posedness of the equilibrium equation even if the width $m$ is only square of the sample size $n$.

## 6 CONCLUSION AND FUTURE WORK

In this paper, we provided a convergence theory for implicit neural networks with ReLU activation in the over-parameterization regime. We showed that the random initialized gradient descent method with fixed step-size converges to a global minimum of the loss function at a linear rate if the width $m = \tilde{\Omega}(n^2)$. In particular, by using a fixed scalar $\gamma \in (0, 1)$ to scale the random initialized weight matrix $\boldsymbol{A}$, we proved that the equilibrium equation is always well-posed throughout the training. By analyzing the gradient flow, we observe that the dynamics of the prediction vector is controlled by a Gram matrix whose smallest eigenvalue is lower bounded by a strictly positive constant as long as $m = \tilde{\Omega}(n^2)$. We envision several potential future directions based on the observations made in the experiments. First, we believe that our analysis can be generalized to implicit neural networks with other scaling techniques and initialization. Here, we use a scalar $\gamma/\sqrt{m}$ to ensure the existence of the equilibrium point $\boldsymbol{z}^*$ with random initialization. With an appropriate normalization, the global convergence for identity initialization can be obtained. Second, we believe that the width $m$ not only improves the convergence rates but also the generalization performance. In particular, our experimental results showed that with a larger $m$ value, the test loss is reduced while the classical bell curve of bias-variance trade-off is not observed.

ACKNOWLEDGMENTS

This work has been supported in part by National Science Foundation grants III-2104797, DMS1812666, CAREER CNS-2110259, CNS-2112471, CNS-2102233, CCF-2110252, CNS-21-20448, CCF-19-34884 and a Google Faculty Research Award.

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

# A APPENDIX

## A.1 PROOF OF LEMMA 2.2

*Proof.* Let $\gamma_0 \in (0, 1)$. Set $\gamma \triangleq \min\{\gamma_0, \gamma_0/c\}$. Denote $\tilde{\gamma} = \gamma/\sqrt{m}$. Then

$$
\begin{aligned}
\|\boldsymbol{z}^{\ell+1} - \boldsymbol{z}^\ell\| &= \|\sigma\left(\tilde{\gamma}\boldsymbol{A}\boldsymbol{z}^\ell + \boldsymbol{\phi}\right) - \sigma\left(\tilde{\gamma}\boldsymbol{A}\boldsymbol{z}^{\ell-1} + \boldsymbol{\phi}\right)\| \\
&\leq \tilde{\gamma}\|\boldsymbol{A}\boldsymbol{z}^\ell - \boldsymbol{A}\boldsymbol{z}^{\ell-1}\|, \quad \sigma \text{ is 1-Lipschitz continuous} \\
&= \tilde{\gamma}\|\boldsymbol{A}(\boldsymbol{z}^\ell - \boldsymbol{z}^{\ell-1})\| \\
&\leq \tilde{\gamma}\|\boldsymbol{A}\|\|\boldsymbol{z}^\ell - \boldsymbol{z}^{\ell-1}\|, \\
&\leq \tilde{\gamma}c\sqrt{m}\|\boldsymbol{z}^\ell - \boldsymbol{z}^{\ell-1}\|, \quad \|\boldsymbol{A}\| \leq c\sqrt{m} \\
&= \gamma_0\|\boldsymbol{z}^\ell - \boldsymbol{z}^{\ell-1}\|.
\end{aligned}
$$

Applying the above argument $\ell$ times, we obtain

$$
\|\boldsymbol{z}^{\ell+1} - \boldsymbol{z}^\ell\| \leq \gamma_0^\ell\|\boldsymbol{z}^1 - \boldsymbol{z}^0\| = \gamma_0^\ell\|\boldsymbol{z}^1\| = \gamma_0^\ell\|\sigma(\boldsymbol{\phi})\| \leq \gamma_0^\ell\|\boldsymbol{\phi}\|,
$$

where we use the fact $\boldsymbol{z}^0 = \boldsymbol{0}$. For any positive integers $p, q$ with $p \leq q$, we have

$$
\begin{aligned}
\|\boldsymbol{z}^p - \boldsymbol{z}^q\| &\leq \|\boldsymbol{z}^p - \boldsymbol{z}^{p+1}\| + \cdots + \|\boldsymbol{z}^{q-1} - \boldsymbol{z}^q\| \\
&\leq \gamma_0^p\|\boldsymbol{\phi}\| + \cdots + \gamma_0^q\|\boldsymbol{\phi}\| \\
&\leq \gamma_0^p\|\boldsymbol{\phi}\|\left(1 + \gamma_0 + \gamma_0^2 + \cdots\right) \\
&= \frac{\gamma_0^p}{1 - \gamma_0}\|\boldsymbol{\phi}\|.
\end{aligned}
$$

Since $\gamma_0 \in (0, 1)$, we have $\|\boldsymbol{z}^p - \boldsymbol{z}^q\| \to 0$ as $p \to \infty$. Hence, $\{\boldsymbol{z}^\ell\}_{\ell=1}^\infty$ is a Cauchy sequence. Since $\mathbb{R}^m$ is complete, the equilibrium point $\boldsymbol{z}^*$ is the limit of the sequence $\{\boldsymbol{z}^\ell\}_{\ell=1}^\infty$, so that $\boldsymbol{z}$ exists and is unique. Moreover, let $q \to \infty$, then we obtain $\|\boldsymbol{z}^p - \boldsymbol{z}^*\| \leq \frac{\gamma^p}{1-\gamma}\|\boldsymbol{\phi}\|$, so that the fixed-point iteration converges to $\boldsymbol{z}$ linearly.

Let $p = 0$ and $q = \ell$, then we obtain $\|\boldsymbol{z}^\ell\| \leq \frac{1}{1-\gamma_0}\|\boldsymbol{\phi}\|$.

$\square$

## A.2 PROOF OF LEMMA 2.3

*Proof.* (i) To simplify the notations, we denote $\boldsymbol{D} \triangleq \mathbf{diag}(\sigma'(\tilde{\gamma}\boldsymbol{A}\boldsymbol{z} + \boldsymbol{\phi}))$, and $\boldsymbol{E} \triangleq \mathbf{diag}(\sigma'(\boldsymbol{W}\boldsymbol{x}))$. The differential of $f$ is given by

$$
\begin{aligned}
df &= d(\boldsymbol{z} - \tilde{\gamma}\sigma(\tilde{\gamma}\boldsymbol{A}\boldsymbol{z} + \boldsymbol{\phi})) \\
&= d\boldsymbol{z} - \boldsymbol{D}d(\tilde{\gamma}\boldsymbol{A}\boldsymbol{z} + \boldsymbol{\phi}) \\
&= [I_m - \tilde{\gamma}\boldsymbol{D}\boldsymbol{A}]\,d\boldsymbol{z} - \tilde{\gamma}\boldsymbol{D}(d\boldsymbol{A})\boldsymbol{z} - \boldsymbol{D}d\boldsymbol{\phi}.
\end{aligned}
$$

Taking vectorization on both sides yields

$$
\begin{aligned}
\mathrm{vec}\,(df) &= [\boldsymbol{I}_m - \tilde{\gamma}\boldsymbol{D}\boldsymbol{A}]\,\mathrm{vec}\,(d\boldsymbol{z}) - \mathrm{vec}\,(\tilde{\gamma}\boldsymbol{D}d\boldsymbol{A}\boldsymbol{z}) - \boldsymbol{D}\mathrm{vec}\,(d\boldsymbol{\phi}) \\
&= [\boldsymbol{I}_m - \tilde{\gamma}\boldsymbol{D}\boldsymbol{A}]\,\mathrm{vec}\,(d\boldsymbol{z}) - \tilde{\gamma}[\boldsymbol{z}^T \otimes \boldsymbol{D}]\mathrm{vec}\,(d\boldsymbol{A}) - \boldsymbol{D}\mathrm{vec}\,(d\boldsymbol{\phi})\,.
\end{aligned}
$$

Therefore, the partial derivative of $f$ with respect to $\boldsymbol{z}$, $\boldsymbol{A}$, and $\boldsymbol{\phi}$ are given by

$$
\begin{aligned}
\frac{\partial f}{\partial \boldsymbol{z}} &= [\boldsymbol{I}_m - \tilde{\gamma}\boldsymbol{D}\boldsymbol{A}]^T, \\
\frac{\partial f}{\partial \boldsymbol{A}} &= -\tilde{\gamma}\left[\boldsymbol{z}^T \otimes \boldsymbol{D}\right]^T, \\
\frac{\partial f}{\partial \boldsymbol{\phi}} &= -\boldsymbol{D}^T.
\end{aligned}
$$

It follows from the definition of the feature vector $\phi$ in Eq. (2) that

$$d\phi = \frac{1}{\sqrt{m}}d\sigma(\boldsymbol{W}\boldsymbol{x}) = \frac{1}{\sqrt{m}}\boldsymbol{E}(d\boldsymbol{W})\boldsymbol{x} = \frac{1}{\sqrt{m}}\left[\boldsymbol{x}^T \otimes \boldsymbol{E}\right]\operatorname{vec}(d\boldsymbol{W}).$$

Thus, the partial derivative of $\phi$ with respect to $\boldsymbol{W}$ is given by

$$\frac{\partial\phi}{\partial\boldsymbol{W}} = \frac{1}{\sqrt{m}}\left[\boldsymbol{x}^T \otimes \boldsymbol{E}\right]^T. \tag{18}$$

By using the chain rule, we obtain the partial derivative of $f$ with respect to $\boldsymbol{W}$ as follows

$$\frac{\partial f}{\partial\boldsymbol{W}} = \frac{\partial\phi}{\partial\boldsymbol{W}}\frac{\partial f}{\partial\phi} = -\frac{1}{\sqrt{m}}\left[\boldsymbol{x}^T \otimes \boldsymbol{E}\right]^T \boldsymbol{D}^T.$$

(ii) Let $\boldsymbol{v}$ be an arbitrary vector, and $\boldsymbol{u}$ be an arbitrary unit vector. The reverse triangle inequality implies that

$$\begin{aligned}
\|(\boldsymbol{I}_m - \tilde{\gamma}\operatorname{diag}(\sigma'(\boldsymbol{v}))\boldsymbol{A})\boldsymbol{u}\| &\geq \|\boldsymbol{u}\| - \|\tilde{\gamma}\operatorname{diag}(\sigma'(\boldsymbol{v}))\boldsymbol{A}\boldsymbol{u}\| \\
&\geq \|\boldsymbol{u}\| - \tilde{\gamma}\|\operatorname{diag}(\sigma'(\boldsymbol{v}))\|\|\boldsymbol{A}\|\|\boldsymbol{u}\| \\
&\overset{(a)}{\geq} (1 - \gamma_0)\|\boldsymbol{u}\| \\
&= 1 - \gamma_0 > 0,
\end{aligned}$$

where $(a)$ is due to $|\sigma'(v)| \leq 1$ and $\|\boldsymbol{A}\|_{\operatorname{op}} \leq c\sqrt{m}$. Therefore, taking infimum on the left-hand side over all unit vector $\boldsymbol{u}$ yields the desired result.

(iii) Since $f(\boldsymbol{z}^*, \boldsymbol{A}, \boldsymbol{W}) = 0$, taking implicit differentiation of $f$ with respect to $\boldsymbol{A}$ at $\boldsymbol{z}^*$ gives us

$$\left(\left.\frac{\partial\boldsymbol{z}}{\partial\boldsymbol{A}}\right|_{z=z^*}\right)\left(\left.\frac{\partial f}{\partial\boldsymbol{z}}\right|_{z=z^*}\right) + \left(\left.\frac{\partial f}{\partial\boldsymbol{A}}\right|_{z=z^*}\right) = 0.$$

The results in part (i)-(ii) imply the smallest eigenvalue of $\left.\frac{\partial f}{\partial\boldsymbol{z}}\right|_{\boldsymbol{z}^*}$ is strictly positive, so that it is invertible. Therefore, we have

$$\frac{\partial\boldsymbol{z}^*}{\partial\boldsymbol{A}} = -\left(\left.\frac{\partial f}{\partial\boldsymbol{A}}\right|_{z=z^*}\right)\left(\left.\frac{\partial f}{\partial\boldsymbol{z}}\right|_{z=z^*}\right)^{-1} = \tilde{\gamma}\left[\boldsymbol{z}^T \otimes \boldsymbol{D}\right]^T\left[\boldsymbol{I}_m - \tilde{\gamma}\boldsymbol{D}\boldsymbol{A}\right]^{-T}. \tag{19}$$

Similarly, we obtain the partial derivative of $\boldsymbol{z}^*$ with respect to $\boldsymbol{W}$ as follows

$$\frac{\partial\boldsymbol{z}^*}{\partial\boldsymbol{W}} = -\left(\left.\frac{\partial f}{\partial\boldsymbol{W}}\right|_{z=z^*}\right)\left(\left.\frac{\partial f}{\partial\boldsymbol{z}}\right|_{z=z^*}\right)^{-1} = \frac{1}{\sqrt{m}}\left[\boldsymbol{x}^T \otimes \boldsymbol{E}\right]^T \boldsymbol{D}^T\left[\boldsymbol{I}_m - \tilde{\gamma}\boldsymbol{D}\boldsymbol{A}\right]^{-T}. \tag{20}$$

To further simplify the notation, we denote $\boldsymbol{z}$ to be the equilibrium point $\boldsymbol{z}^*$ by omitting the superscribe, i.e., $\boldsymbol{z} = \boldsymbol{z}^*$. Let $\hat{y} = \boldsymbol{u}^T\boldsymbol{z} + \boldsymbol{v}^T\phi$ be the prediction for the training data $(\boldsymbol{x}, \boldsymbol{y})$. The differential of $\hat{y}$ is given by

$$d\hat{y} = d\left(\boldsymbol{u}^T\boldsymbol{z} + \boldsymbol{v}^T\phi\right) = \boldsymbol{u}^T d\boldsymbol{z} + \boldsymbol{z}d\boldsymbol{u} + \boldsymbol{v}^T d\phi + \phi^T d\boldsymbol{v}.$$

The partial derivative of $\hat{y}$ with respect to $\boldsymbol{u}$, $\boldsymbol{v}$, $\boldsymbol{z}$, and $\phi$ are given by

$$\frac{\partial\hat{y}}{\partial\boldsymbol{z}} = \boldsymbol{u}, \quad \frac{\partial\hat{y}}{\partial\boldsymbol{u}} = \boldsymbol{z}, \quad \frac{\partial\hat{y}}{\partial\boldsymbol{v}} = \phi, \quad \frac{\partial\hat{y}}{\partial\phi} = \boldsymbol{v}. \tag{21}$$

Let $\ell = \frac{1}{2}(\hat{y} - y)^2$. Then $\partial\ell/\partial\hat{y} = (\hat{y} - y)$. By chain rule, we have

$$\frac{\partial\ell}{\partial\boldsymbol{u}} = \frac{\partial\hat{y}}{\partial\boldsymbol{u}}\frac{\partial\ell}{\partial\hat{y}} = \boldsymbol{z}(\hat{y} - y), \tag{22}$$

$$\frac{\partial\ell}{\partial\phi} = \frac{\partial\hat{y}}{\partial\boldsymbol{v}}\frac{\partial\ell}{\partial\hat{y}} = \phi(\hat{y} - y). \tag{23}$$

By using (19)-(20) and chain rule, we obtain

$$\begin{aligned}
\frac{\partial\ell}{\partial\boldsymbol{A}} &= \frac{\partial\boldsymbol{z}}{\partial\boldsymbol{A}}\frac{\partial\ell}{\partial\boldsymbol{z}} \\
&= \frac{\partial\boldsymbol{z}}{\partial\boldsymbol{A}}\frac{\partial\hat{y}}{\partial\boldsymbol{z}}\frac{\partial\ell}{\partial\hat{y}} = \tilde{\gamma}(\hat{y} - y)\left[\boldsymbol{z}^T \otimes \boldsymbol{D}\right]^T\left[\boldsymbol{I}_m - \tilde{\gamma}\boldsymbol{D}\boldsymbol{A}\right]^{-T}\boldsymbol{u}, \tag{24}
\end{aligned}$$

and

$$\frac{\partial \ell}{\partial \boldsymbol{W}} = \frac{\partial \boldsymbol{z}}{\partial \boldsymbol{W}} \frac{\partial \hat{y}}{\partial \boldsymbol{z}} \frac{\partial \ell}{\partial \hat{y}} + \frac{\partial \boldsymbol{\phi}}{\partial \boldsymbol{W}} \frac{\partial \hat{y}}{\partial \boldsymbol{\phi}} \frac{\partial \ell}{\partial \hat{y}}$$

$$= \frac{1}{\sqrt{m}} (\hat{y} - y)[\boldsymbol{x}^T \otimes \boldsymbol{E}]^T \left[ \boldsymbol{D}^T (I_m - \tilde{\gamma} \boldsymbol{D} \boldsymbol{A})^{-T} \boldsymbol{u} + \boldsymbol{v} \right]. \tag{25}$$

Since $L = \sum_{i=1}^n \ell_i$ with $\ell_i = \ell(\hat{y}_i, y_i)$, we have $dL = \sum_{i=1}^n d\ell_i$ and $\partial L / \partial \ell_i = 1$. Therefore, we obtain

$$\frac{\partial L}{\partial \boldsymbol{A}} = \sum_{i=1}^n \frac{\partial \ell_i}{\partial \boldsymbol{A}} = \sum_{i=1}^n \tilde{\gamma} (\hat{y}_i - y_i) \left[ \boldsymbol{z}_i^T \otimes \boldsymbol{D}_i \right]^T \left[ I_m - \tilde{\gamma} \boldsymbol{D}_i \boldsymbol{A} \right]^{-T} \boldsymbol{u}, \tag{26}$$

$$\frac{\partial L}{\partial \boldsymbol{W}} = \sum_{i=1}^n \frac{\partial \ell_i}{\partial \boldsymbol{W}} = \sum_{i=1}^n \frac{1}{\sqrt{m}} (\hat{y}_i - y_i)[\boldsymbol{x}_i^T \otimes \boldsymbol{E}_i]^T \left[ \boldsymbol{D}_i^T (I_m - \tilde{\gamma} \boldsymbol{D}_i \boldsymbol{A})^{-T} \boldsymbol{u} + \boldsymbol{v} \right], \tag{27}$$

$$\frac{\partial L}{\partial \boldsymbol{u}} = \sum_{i=1}^n \frac{\partial \ell_i}{\partial \boldsymbol{u}} = \sum_{i=1}^n (\hat{y}_i - y_i) \boldsymbol{z}_i, \tag{28}$$

$$\frac{\partial L}{\partial \boldsymbol{v}} = \sum_{i=1}^n \frac{\partial \ell_i}{\partial \boldsymbol{v}} = \sum_{i=1}^n (\hat{y}_i - y_i) \boldsymbol{\phi}_i. \tag{29}$$

$\square$

### A.3 PROOF OF LEMMA 3.1

*Proof.* Let $\boldsymbol{z}_i$ denote the $i$-th equilibrium point for the $i$-the data sample $\boldsymbol{x}_i$. By using (19), (20), (26) and (27), we obtain the dynamics of the equilibrium point $\boldsymbol{z}_i$ as follows

$$\frac{d\boldsymbol{z}_i}{dt} = \left( \frac{\partial \boldsymbol{z}_i}{\partial \boldsymbol{A}} \right)^T \frac{d\text{vec}\,(\boldsymbol{A})}{dt} + \left( \frac{\partial \boldsymbol{z}_i}{\partial \boldsymbol{W}} \right)^T \frac{d\text{vec}\,(\boldsymbol{W})}{dt}$$

$$= \left( \frac{\partial \boldsymbol{z}_i}{\partial \boldsymbol{A}} \right)^T \left( -\frac{\partial L}{\partial \boldsymbol{A}} \right) + \left( \frac{\partial \boldsymbol{z}_i}{\partial \boldsymbol{W}} \right)^T \left( -\frac{\partial L}{\partial \boldsymbol{W}} \right)$$

$$= -\tilde{\gamma}^2 \sum_{j=1}^n (\hat{y}_j - y_j) \left[ I_m - \tilde{\gamma} \boldsymbol{D}_i \boldsymbol{A} \right]^{-1} \left[ \boldsymbol{z}_i^T \otimes \boldsymbol{D}_i \right] \left[ \boldsymbol{z}_j^T \otimes \boldsymbol{D}_j \right]^T \left[ I_m - \tilde{\gamma} \boldsymbol{D}_j \boldsymbol{A} \right]^{-T} \boldsymbol{u}$$

$$- \frac{1}{m} \sum_{j=1}^n (\hat{y}_j - y_j) \left[ I_m - \tilde{\gamma} \boldsymbol{D}_i \boldsymbol{A} \right]^{-1} \boldsymbol{D}_i \left[ \boldsymbol{x}_i^T \otimes \boldsymbol{E}_i \right] \left[ \boldsymbol{x}_j^T \otimes \boldsymbol{E}_j \right]^T \left[ \boldsymbol{D}_j^T (I_m - \tilde{\gamma} \boldsymbol{D}_j \boldsymbol{A})^{-T} \boldsymbol{u} + \boldsymbol{v} \right]$$

$$= -\tilde{\gamma}^2 \sum_{j=1}^n (\hat{y}_j - y_j) \left[ I_m - \tilde{\gamma} \boldsymbol{D}_i \boldsymbol{A} \right]^{-1} \boldsymbol{D}_i \boldsymbol{D}_j^T \left[ I_m - \tilde{\gamma} \boldsymbol{D}_j \boldsymbol{A} \right]^{-T} \boldsymbol{u} \boldsymbol{z}_i^T \boldsymbol{z}_j$$

$$- \frac{1}{m} \sum_{j=1}^n (\hat{y}_j - y_j) \left[ I_m - \tilde{\gamma} \boldsymbol{D}_i \boldsymbol{A} \right]^{-1} \boldsymbol{D}_i \boldsymbol{E}_i \boldsymbol{E}_j^T \left[ \boldsymbol{D}_j^T (I_m - \tilde{\gamma} \boldsymbol{D}_j \boldsymbol{A})^{-T} \boldsymbol{u} + \boldsymbol{v} \right] \boldsymbol{x}_i^T \boldsymbol{x}_j.$$

By using (18) and 27, we obtain the dynamics of the feature vector $\boldsymbol{\phi}_i$

$$\frac{d\boldsymbol{\phi}_i}{dt} = \left( \frac{\partial \boldsymbol{\phi}_i}{\partial \boldsymbol{W}} \right)^T \frac{d\text{vec}\,(\boldsymbol{W})}{dt}$$

$$= \left( \frac{\partial \boldsymbol{\phi}_i}{\partial \boldsymbol{W}} \right)^T \left( -\frac{\partial L}{\partial W} \right)$$

$$= -\frac{1}{m} \sum_{j=1}^n (\hat{y}_i - y_i) \boldsymbol{E}_i \boldsymbol{E}_j^T [\boldsymbol{D}_j^T (I_m - \tilde{\gamma} \boldsymbol{D}_j \boldsymbol{A})^{-T} \boldsymbol{u} + \boldsymbol{v}] \boldsymbol{x}_i^T \boldsymbol{x}_j.$$

By chain rule, the dynamics of the prediction $\hat{y}_i$ is given by

$$
\frac{d\hat{y}_i}{dt} = \left(\frac{\partial \hat{y}_i}{\partial \boldsymbol{z}_i}\right)^T \frac{d\boldsymbol{z}_i}{dt} + \left(\frac{\partial \hat{y}_i}{\partial \boldsymbol{\phi}_i}\right)^T \frac{d\boldsymbol{\phi}_i}{dt} + \left(\frac{\partial \hat{y}_i}{\partial \boldsymbol{u}}\right)^T \frac{d\boldsymbol{u}}{dt} + \left(\frac{\partial \hat{y}_i}{\partial \boldsymbol{v}}\right)^T \frac{d\boldsymbol{v}}{dt}
$$

$$
= -\tilde{\gamma}^2 \sum_{j=1}^n (\hat{y}_j - y_j) \left[\boldsymbol{u}^T (I_m - \tilde{\gamma}\boldsymbol{D}_i\boldsymbol{A})^{-1} \boldsymbol{D}_i \boldsymbol{D}_j^T (I_m - \tilde{\gamma}\boldsymbol{D}_j\boldsymbol{A})^{-T}\boldsymbol{u}\right] (\boldsymbol{z}_i^T \boldsymbol{z}_j)
$$

$$
- \frac{1}{m} \sum_{j=1}^n (\hat{y}_j - y_j) \left[\left(\boldsymbol{D}_i^T (I_m - \tilde{\gamma}\boldsymbol{D}_i\boldsymbol{A})^{-1}\boldsymbol{u} + \boldsymbol{v}\right)^T \boldsymbol{E}_i \boldsymbol{E}_j^T \left(\boldsymbol{D}_j^T (I_m - \tilde{\gamma}\boldsymbol{D}_j\boldsymbol{A})^{-T}\boldsymbol{u} + \boldsymbol{v}\right)\right] (\boldsymbol{x}_i^T \boldsymbol{x}_j)
$$

$$
- \sum_{j=1}^n (\hat{y}_j - y_j)(\boldsymbol{z}_i^T \boldsymbol{z}_j)
$$

$$
- \sum_{j=1}^n (\hat{y}_j - y_j)(\boldsymbol{\phi}_i^T \boldsymbol{\phi}_j).
$$

Define the matrices $\boldsymbol{M}(t) \in \mathbb{R}^{n \times n}$ and $\boldsymbol{Q}(t) \in \mathbb{R}^{n \times n}$ as follows

$$
\boldsymbol{M}(t)_{ij} \triangleq \frac{1}{m}\boldsymbol{u}^T (I_m - \tilde{\gamma}\boldsymbol{D}_i\boldsymbol{A})^{-1}\boldsymbol{D}_i\boldsymbol{D}_j^T (I_m - \tilde{\gamma}\boldsymbol{D}_j\boldsymbol{A})^{-T}\boldsymbol{u},
$$

$$
\boldsymbol{Q}(t)_{ij} \triangleq \frac{1}{m} \left(\boldsymbol{D}_i^T (I_m - \tilde{\gamma}\boldsymbol{D}_i\boldsymbol{A})^{-1}\boldsymbol{u} + \boldsymbol{v}\right)^T \boldsymbol{E}_i\boldsymbol{E}_j^T \left(\boldsymbol{D}_j^T (I_m - \tilde{\gamma}\boldsymbol{D}_j\boldsymbol{A})^{-T}\boldsymbol{u} + \boldsymbol{v}\right).
$$

Let $\boldsymbol{X} \in \mathbb{R}^{n \times d}$, $\boldsymbol{\Phi}(t) \in \mathbb{R}^{n \times m}$, and $\boldsymbol{Z}(t) \in \mathbb{R}^{n \times m}$ be the matrices whose rows are the training data $\boldsymbol{x}_i$, feature vectors $\boldsymbol{\phi}_i$, and equilibrium points $\boldsymbol{z}_i$ at time $t$, respectively. The dynamics of the prediction vector $\hat{\boldsymbol{y}}$ is given by

$$
\frac{d\hat{\boldsymbol{y}}}{dt} = -\left[\left(\gamma^2 \boldsymbol{M}(t) + \boldsymbol{I}_n\right) \circ \boldsymbol{Z}(t)\boldsymbol{Z}(t)^T + \boldsymbol{Q}(t) \circ \boldsymbol{X}\boldsymbol{X}^T + \boldsymbol{\Phi}(t)\boldsymbol{\Phi}(t)^T\right] (\hat{\boldsymbol{y}}(t) - \boldsymbol{y}).
$$

$\square$

## A.4  PROOF OF LEMMA 3.2

### A.4.1  REVIEW OF HERMITE EXPANSIONS

To make the paper self-contained, we review the necessary background about the Hermite polynomials in this section. One can find each result in this section from any standard textbooks about functional analysis such as MacCluer (2008); Kreyszig (1978), or most recent literature (Nguyen & Mondelli, 2020, Appendix D) and (Oymak & Soltanolkotabi, 2020, Appendix H).

We consider an $L^2$-space defined by $L^2(\mathbb{R}, dP)$, where $dP$ is the *Gaussian measure*, that is,

$$
dP = p(x)dx, \quad \text{where} \quad p(x) = \frac{1}{\sqrt{2\pi}}e^{-\frac{x^2}{2}}.
$$

Thus, $L^2(\mathbb{R}, dP)$ is a collection of functions $f$ for which

$$
\int_{-\infty}^{\infty} |f(x)|^2 \, dP(x) = \int_{-\infty}^{\infty} |f(x)|^2 \, p(x)dx = \mathbb{E}_{x \sim N(0,1)} |f(x)|^2 < \infty.
$$

**Lemma A.1.** The ReLU activation $\sigma \in L^2(\mathbb{R}, dP)$.

*Proof.* Note that

$$
\int_{-\infty}^{\infty} |\sigma(x)|^2 \, p(x)dx \le \int_{-\infty}^{\infty} |x|^2 \, p(x)dx = \mathbb{E}_{x \sim N(0,1)} |x|^2 = \text{Var}(x) = 1.
$$

$\square$

For any functions $f, g \in L^2(\mathbb{R}, dP)$, we define an *inner product*

$$\langle f, g \rangle := \int_{-\infty}^{\infty} f(x)g(x)dP(x) = \int_{-\infty}^{\infty} f(x)g(x)p(x)dx = \mathbb{E}_{x \sim N(0,1)}[f(x)g(x)].$$

Furthermore, the induced norm $\| \cdot \|$ is given by

$$\|f\|^2 = \langle f, f \rangle = \int_{-\infty}^{\infty} |f(x)|^2 \, dP(x) = \mathbb{E}_{x \sim N(0,1)} |f(x)|^2 .$$

This $L^2$ space has an orthonormal basis with respect to the inner product defined above, called *normalized probabilist's Hermite polynomials* $\{h_n(x)\}_{n=0}^{\infty}$ that are given by

$$h_n(x) = \frac{1}{\sqrt{n!}}(-1)^n e^{x^2/2} D^n(e^{-x^2/2}), \quad \text{where} \quad D^n(e^{-x^2/2}) = \frac{d^n}{dx^n} e^{-x^2/2}.$$

**Lemma A.2.** The *normalized probabilist's Hermite polynomials* is an orthonormal basis of $L^2(\mathbb{R}, dP)$: $\langle h_m, h_n \rangle = \delta_{mn}$.

*Proof.* Note that $D^n(e^{-x^2/2}) = e^{-x^2/2}P_n(x)$ for a polynomial with degree of $n$ and leading term is $(-1)^n x^n$. Thus, we can consider $h_n(x) = \frac{1}{\sqrt{n!}}(-1)^n P_n(x)$.

Assume $m < n$

$$
\begin{aligned}
\langle h_n, h_m \rangle &= \mathbb{E}_{x \sim N(0,1)}[h_n(x)h_m(x)] \\
&= \int_{-\infty}^{\infty} h_n(x)h_m(x)\frac{1}{\sqrt{2\pi}}e^{-x^2/2}dx, \\
&= \frac{1}{\sqrt{2\pi}\sqrt{n!}}(-1)^n \int_{-\infty}^{\infty} D^n(e^{-x^2/2})h_m(x)dx, \quad \text{rewrite } h_n(x) \text{ by its definition} \\
&= \frac{1}{\sqrt{2\pi}\sqrt{n!}\sqrt{m!}}(-1)^{n+m} \int_{-\infty}^{\infty} D^n(e^{-x^2/2})P_m(x)dx, \quad \text{rewrite } h_m \text{ by the polynomial form} \\
&= \frac{1}{\sqrt{2\pi}\sqrt{n!}\sqrt{m!}}(-1)^{2n+m} \int_{-\infty}^{\infty} e^{-x^2/2}D_n[P_m(x)]dx, \quad \text{integration by parts } n \text{ times}
\end{aligned}
$$

There is no boundary terms because the super exponential decay of $e^{-x^2/2}$ at infinity. Since $m < n$, then $D_n(P_m) = 0$ so that $\langle h_m, h_n \rangle = 0$. If $m = n$, then $D_n(P_m) = (-1)^n n!$. Thus, $\langle h_n, h_n \rangle = 1$. $\qquad \square$

**Remark**: Since $\{h_n\}$ is an orthonormal basis, for every $f \in L^2(\mathbb{R}, dP)$, we have

$$f(x) = \sum_{n=0}^{\infty} \langle f, h_n \rangle h_n(x)$$

in the sense that

$$\lim_{N \to \infty} \left\| f(x) - \sum_{n=0}^{N} \langle f, h_n \rangle h_n(x) \right\|^2 = \lim_{N \to \infty} \mathbb{E}_{x \sim N(0,1)} \left| f(x) - \sum_{n=0}^{N} \langle f, h_n \rangle h_n(x) \right|^2 = 0$$

**Lemma A.3.** $f \in L^2(\mathbb{R}, dP)$ if and only if $\sum_{n=0}^{\infty} |\langle f, h_n \rangle|^2 < \infty$.

*Proof.* Note that

$$\langle f, f \rangle = \int_{-\infty}^{\infty} |f(x)|^2 \, dP(x)$$

$$= \int_{-\infty}^{\infty} \left( \sum_{i=0}^{\infty} \langle f, h_i \rangle \, h_i(x) \right) \left( \sum_{j=0}^{\infty} \langle f, h_j \rangle \, h_j(x) \right) dP(x)$$

$$= \sum_{i,j=0}^{\infty} \langle f, h_i \rangle \langle f, h_j \rangle \int_{-\infty}^{\infty} h_i(x) h_j(x) dP(x)$$

$$= \sum_{i=1}^{\infty} |\langle f, h_i \rangle|^2 .$$

$\square$

**Lemma A.4.** Consider a Hilbert space $H$ with inner product $\langle \cdot, \cdot \rangle$. If $\|f_n - f\| \to 0$ and $\|g_n - g\| \to 0$, then $\langle f, g \rangle = \lim_{n \to \infty} \langle f_n, g_n \rangle$.

*Proof.* Observe that

$$|\langle f, g \rangle - \langle f_n, g_n \rangle| \leq |\langle f, g \rangle - \langle f_n, g \rangle| + |\langle f_n, g \rangle - \langle f_n, g_n \rangle|$$
$$\leq \|f\| \|g - g_n\| + \|f_n\| \|g - g_n\|.$$

Let $n \to \infty$, then the continuity of $\| \cdot \|$ implies the desired result. $\square$

**Lemma A.5.** Let $\{h_n(x)\}$ be the normalized probabilist's Hermite polynomials. For any fixed number $t$, we have

$$e^{xt - t^2/2} = \sum_{n=0}^{\infty} \frac{t^n}{\sqrt{n!}} h_n(x). \tag{30}$$

*Proof.* First, we show $f(x) = e^{xt - t^2/2} \in H \triangleq L^2(\mathbb{R}, dP)$.

$$\langle f, f \rangle = \mathbb{E}_{x \sim N(0,1)} |f(x)|^2$$

$$= \int_{-\infty}^{\infty} e^{2xt - t^2} \frac{1}{\sqrt{2\pi}} e^{-x^2/2} dx$$

$$= e^{t^2} \int_{-\infty}^{\infty} \frac{1}{\sqrt{2\pi}} \exp\left\{ -\frac{(x - 2t)^2}{2} \right\} dx, \quad x \sim N(2t, 1)$$

$$= e^{t^2} < \infty.$$

Thus $f(x) \in H$. Then $f(x) = \sum_{n=0}^{\infty} \langle f, h_n \rangle h_n(x)$. Note that

$$\langle f, h_n \rangle = \mathbb{E}_{x \sim N(0,1)}[f(x) h_n(x)]$$

$$= \int_{-\infty}^{\infty} e^{xt - t^2/2} \cdot \frac{1}{\sqrt{n!}} (-1)^n e^{x^2/2} D_n(e^{-x^2/2}) \cdot \frac{1}{\sqrt{2\pi}} e^{-x^2/2} dx$$

$$= \frac{1}{\sqrt{n!}} (-1)^n \frac{1}{\sqrt{2\pi}} \int_{-\infty}^{\infty} e^{xt - t^2/2} \cdot D_n(e^{-x^2/2}) dx, \quad \text{integration by parts } n \text{ times}$$

$$= \frac{1}{\sqrt{n!}} (-1)^{2n} \frac{1}{\sqrt{2\pi}} \int_{-\infty}^{\infty} e^{xt - t^2/2} t^n \cdot e^{-x^2/2} dx$$

$$= \frac{t^n}{\sqrt{n!}} \int_{-\infty}^{\infty} \frac{1}{\sqrt{2\pi}} e^{-(x-t)^2/2} dx, \quad x \sim N(t, 1)$$

$$= \frac{t^n}{\sqrt{n!}}.$$

$\square$

**Lemma A.6.** Let $\boldsymbol{a}, \boldsymbol{b} \in \mathbb{R}^d$ with $\|\boldsymbol{a}\| = \|\boldsymbol{b}\| = 1$, then

$$\mathbb{E}_{w \sim N(\mathbf{0}, \boldsymbol{I}_d)}[h_n(\langle \boldsymbol{a}, \boldsymbol{w} \rangle) h_m(\langle \boldsymbol{b}, \boldsymbol{w} \rangle)] = \langle \boldsymbol{a}, \boldsymbol{b} \rangle^n \delta_{mn}.$$

*Proof.* Given fixed numbers $s$ and $t$, we define two functions $f(\boldsymbol{w}) = e^{\langle \boldsymbol{a}, \boldsymbol{w} \rangle t - t^2/2}$ and $g(\boldsymbol{w}) = e^{\langle \boldsymbol{b}, \boldsymbol{w} \rangle s - s^2/2}$. Let $x = \langle \boldsymbol{a}, \boldsymbol{w} \rangle$ and $y = \langle \boldsymbol{b}, w \rangle$. Then we have

$$f(\boldsymbol{w}) = e^{\langle \boldsymbol{a}, \boldsymbol{w} \rangle t - t^2/2} = e^{xt - t^2/2} = \sum_{n=0}^{\infty} \frac{t^n}{\sqrt{n!}} h_n(x) = \sum_{n=0}^{\infty} \frac{t^n}{\sqrt{n!}} h_n(\langle \boldsymbol{a}, \boldsymbol{w} \rangle),$$

$$g(\boldsymbol{w}) = e^{\langle \boldsymbol{b}, \boldsymbol{w} \rangle s - s^2/2} = e^{ys - s^2/2} = \sum_{n=0}^{\infty} \frac{s^n}{\sqrt{n!}} h_n(y) = \sum_{n=0}^{\infty} \frac{s^n}{\sqrt{n!}} h_n(\langle \boldsymbol{b}, \boldsymbol{w} \rangle).$$

Define a Hilbert space $H_d = L^2(\mathbb{R}^d, dP)$, where $dP$ is the *multivariate Gaussian measure*, equipped with inner product $\langle f, g \rangle \triangleq \mathbb{E}_{\boldsymbol{w} \sim N(\mathbf{0}, \boldsymbol{I}_d)}[f(\boldsymbol{w}) g(\boldsymbol{w})]$. Clearly, $f, g \in H_d$. Define sequences $\{f_N\}$ and $\{g_N\}$ as follows

$$f_N(\boldsymbol{w}) = \sum_{n=0}^{N} \frac{t^n}{\sqrt{n!}} h_n(\langle \boldsymbol{a}, \boldsymbol{w} \rangle) \quad \text{and} \quad g_N(\boldsymbol{w}) = \sum_{n=0}^{N} \frac{s^n}{\sqrt{n!}} h_n(\langle \boldsymbol{b}, \boldsymbol{w} \rangle).$$

Since $\|f - f_N\| \to 0$ and $\|g - g_N\| \to 0$, we have

$$\begin{aligned}
\mathbb{E}_{\boldsymbol{w} \sim N(0, I_d)}[f(\boldsymbol{w}) g(\boldsymbol{w})] &= \langle f, g \rangle \\
&= \lim_{N \to \infty} \langle f_N, g_N \rangle \\
&= \lim_{N \to \infty} \mathbb{E}_{\boldsymbol{w} \sim N(\mathbf{0}, \boldsymbol{I}_d)}[f_N(\boldsymbol{w}) g_N(\boldsymbol{w})] \\
&= \lim_{N \to \infty} \sum_{n,m=0}^{N} \frac{t^n s^m}{\sqrt{n!} \sqrt{m!}} \mathbb{E}_{\boldsymbol{w} \sim N(\mathbf{0}, \boldsymbol{I}_d)}[h_n(\langle \boldsymbol{a}, \boldsymbol{w} \rangle) g_n(\langle \boldsymbol{b}, \boldsymbol{w} \rangle)]
\end{aligned}$$

Note that the LHS is also given by

$$\begin{aligned}
\mathbb{E}_{\boldsymbol{w} \sim N(\mathbf{0}, \boldsymbol{I}_d)}[f(\boldsymbol{w}) g(\boldsymbol{w})] &= e^{-t^2/2 - s^2/2} \mathbb{E}_{\boldsymbol{w} \sim N(\mathbf{0}, \boldsymbol{I}_d)}[e^{\langle \boldsymbol{a}, \boldsymbol{w} \rangle t + \langle \boldsymbol{b}, \boldsymbol{w} \rangle s}] \\
&= e^{-t^2/2 - s^2/2} \mathbb{E}_{\boldsymbol{w} \sim N(\mathbf{0}, \boldsymbol{I}_d)}[e^{\sum_{i=1}^{d} \boldsymbol{w}_i(a_i t + b_i s)}] \\
&= e^{-t^2/2 - s^2/2} \prod_{i=1}^{d} \mathbb{E}_{w_i \sim N(0,1)}[e^{\boldsymbol{w}_i(a_i t + b_i s)}] \\
&= e^{-t^2/2 - s^2/2} \prod_{i=1}^{d} M_{\boldsymbol{w}_i}(a_i t + b_i s) \\
&= e^{\langle \boldsymbol{a}, \boldsymbol{b} \rangle st} \\
&= \sum_{n=0}^{\infty} \frac{\langle \boldsymbol{a}, \boldsymbol{b} \rangle^n (st)^n}{n!}.
\end{aligned}$$

Since $s$ and $t$ are arbitrary numbers, matching the coefficients yields

$$\mathbb{E}_{\boldsymbol{w} \sim N(0, \boldsymbol{I}_d)}[h_n(\langle a, \boldsymbol{w} \rangle) h_m(\langle b, \boldsymbol{w} \rangle)] = \langle \boldsymbol{a}, \boldsymbol{b} \rangle^n \delta_{mn}.$$

$\square$

### A.4.2 LOWER BOUND THE SMALLEST EIGENVALUES OF $\boldsymbol{G}^\infty$

The result in this subsection is similar to the results in (Nguyen & Mondelli, 2020, Appendix D) and (Oymak & Soltanolkotabi, 2020, Appendix H). The key difference is the assumptions made on the training data. In particular, Oymak & Soltanolkotabi (2020) assumes the training data is $\delta$-separable, *i.e.*, $\min\{\|\boldsymbol{x}_i - \boldsymbol{x}_j\|, \|\boldsymbol{x}_i + \boldsymbol{x}_j\|\} \geq \delta > 0$ for all $i \neq j$, and Nguyen & Mondelli (2020) assumes the data $\boldsymbol{x}_i$ follows some sub-Gaussian random variable, while we assume no two data are parallel to each other, *i.e.*, $\boldsymbol{x}_i \not\parallel \boldsymbol{x}_j$ for all $i \neq j$.

**Lemma A.7.** Given an activation function $\sigma$, if $\sigma \in L^2(\mathbb{R}, dP)$ and $\|\boldsymbol{x}_i\| = 1$ for all $i \in [n]$, then

$$\boldsymbol{G}^\infty = \sum_{k=0}^\infty |\langle \sigma, h_k \rangle|^2 \underbrace{\left( \boldsymbol{X}\boldsymbol{X}^T \circ \cdots \circ \boldsymbol{X}\boldsymbol{X}^T \right)}_{k \text{ times}}, \tag{31}$$

where $\circ$ is elementwise product.

*Proof.* Observe

$$\begin{aligned}
\boldsymbol{G}_{ij}^\infty &= \mathbb{E}_{\boldsymbol{w} \sim N(0, \boldsymbol{I}_d)} \left[ \sigma(\langle \boldsymbol{w}, \boldsymbol{x}_i \rangle) \sigma(\langle \boldsymbol{w}, \boldsymbol{x}_j \rangle) \right] \\
&= \sum_{k, \ell = 0}^\infty \langle \sigma, h_k \rangle \langle \sigma, h_\ell \rangle \, \mathbb{E}_{\boldsymbol{w} \sim N(0, \boldsymbol{I}_d)} \left[ h_k(\langle \boldsymbol{w}, \boldsymbol{x}_i \rangle) h_\ell(\langle \boldsymbol{w}, \boldsymbol{x}_j \rangle) \right] \\
&= \sum_{k, \ell = 0}^\infty \langle \sigma, h_k \rangle \langle \sigma, h_\ell \rangle \cdot \langle \boldsymbol{x}_i, \boldsymbol{x}_j \rangle^k \delta_{k\ell} \\
&= \sum_{k=0}^\infty \langle \sigma, h_k \rangle^2 \langle \boldsymbol{x}_i, \boldsymbol{x}_j \rangle^k
\end{aligned}$$

$\square$

Note that the tensor product of $\boldsymbol{x}_i$ and $\boldsymbol{x}_i$ is $\boldsymbol{x}_i \otimes \boldsymbol{x}_i \in \mathbb{R}^{d^2 \times 1}$, so that

$$\langle \boldsymbol{x}_i, \boldsymbol{x}_j \rangle^k = \left\langle \underbrace{\boldsymbol{x}_i \otimes \cdots \otimes \boldsymbol{x}_i}_{k \text{ times}}, \underbrace{\boldsymbol{x}_j \otimes \cdots \otimes \boldsymbol{x}_j}_{k \text{ times}} \right\rangle$$

Here we introduce the *(row-wise) Khatri–Rao product* of two matrices $\boldsymbol{A} \in \mathbb{R}^{k \times m}$, $\boldsymbol{B} \in \mathbb{R}^{k \times n}$. Then

$$\boldsymbol{A} * \boldsymbol{B} = \begin{bmatrix} \boldsymbol{A}_{1*} \otimes \boldsymbol{B}_{1*} \\ \vdots \\ \boldsymbol{A}_{k*} \otimes \boldsymbol{B}_{k*} \end{bmatrix} \in \mathbb{R}^{k \times mn}$$

where $\boldsymbol{A}_{i*}$ indicates the $i$-th row of matrix $\boldsymbol{A}$. Therefore, the $i$-th row of $\boldsymbol{X} * \cdots * \boldsymbol{X} \triangleq \boldsymbol{X}^{*n}$ is $\boldsymbol{x}_i \otimes \cdots \otimes \boldsymbol{x}_i$. As a result, we obtain a more compact form of (31) as follows

$$\boldsymbol{G}^\infty = \sum_{k=0}^\infty |\langle \sigma, h_k \rangle|^2 (\boldsymbol{X}^{*k})(\boldsymbol{X}^{*k})^T. \tag{32}$$

**Lemma A.8.** If $\sigma(x)$ is a nonlinear function and $|\sigma(x)| \leq |x|$ and , then

$$\sup\{n : \langle \sigma, h_n \rangle > 0\} = \infty.$$

*Proof.* It is equivalent to show $\sigma(x)$ is not a finite linear combination of polynomials. We prove by contradiction. Suppose $\sigma(x) = a_0 + a_1 x + \cdots + a_n x^n$. Since $\sigma(0) = 0 = a_0$, then $\sigma(x) = a_1 x + \cdots + a_n x^n$. Observe that

$$\begin{aligned}
\lim_{x \to \infty} \frac{|\sigma(x)|}{|x|} &= \lim_{x \to \infty} \frac{|a_1 x + \cdots + a_n x^n|}{|x|} \\
&= \lim_{x \to \infty} |a_1 + \cdots + a_n x^{n-1}|, \\
&= \infty
\end{aligned}$$

which contradicts $\frac{|\sigma(x)|}{|x|} \leq 1$ for all $x \neq 0$. $\square$

**Lemma A.9.** If $\boldsymbol{x}_i \not\parallel \boldsymbol{x}_j$ for all $i \neq j$, then there exists $k_0 > 0$ such that $\lambda_{\min}\left[ (\boldsymbol{X}^{*k})(\boldsymbol{X}^{*k})^T \right] > 0$ for all $k \geq k_0$. Therefore, $\lambda_{\min}(\boldsymbol{G}^\infty) > 0$.

*Proof.* To simplify the notation, denote $\boldsymbol{K} = (\boldsymbol{X}^{*k})^T \in \mathbb{R}^{kd \times n}$. Since $x_i \not\parallel \boldsymbol{x}_j$ and $\|\boldsymbol{x}_i\| = 1$, then let $\delta \triangleq \max\{|\langle \boldsymbol{x}_i, \boldsymbol{x}_j \rangle|\} = \max\{|\cos \theta_{ij}|\}$ and $\delta \in (0, 1)$, where $\theta_{ij}$ is the angle between $\boldsymbol{x}_i$ and $\boldsymbol{x}_j$. For any unit vector $\boldsymbol{v} \in \mathbb{R}^n$, we have

$$\boldsymbol{v}^T (\boldsymbol{X}^{*k})(\boldsymbol{X}^{*k})^T \boldsymbol{v} = \|\boldsymbol{K}\boldsymbol{v}\|^2 = \left\| \sum_{i=1}^n v_i \boldsymbol{K}_{*i} \right\|^2$$

$$= \sum_{i=1}^n \sum_{j=1}^n v_i v_j \langle \boldsymbol{K}_{*i}, \boldsymbol{K}_{*j} \rangle$$

$$= \sum_{i=1}^n \sum_{j=1}^n v_i v_j \langle \boldsymbol{x}_i, \boldsymbol{x}_j \rangle^k$$

$$= \sum_{i=1}^n v_i^2 \|\boldsymbol{x}_i\|^{2k} + \sum_{i \neq j} v_i v_j \langle \boldsymbol{x}_i, \boldsymbol{x}_j \rangle^k$$

$$= 1 + \sum_{i \neq j} v_i v_j \langle \boldsymbol{x}_i, \boldsymbol{x}_j \rangle^k,$$

where the last equality is because $\|\boldsymbol{x}_i\| = 1$ and $\|\boldsymbol{v}\| = 1$. Note that

$$\left| \sum_{i \neq j} v_i v_j \langle \boldsymbol{x}_i, \boldsymbol{x}_j \rangle^k \right| \leq \sum_{i \neq j} |v_i| \, |v_j| \, |\langle \boldsymbol{x}_i, \boldsymbol{x}_j \rangle|^k$$

$$\leq \delta^k \sum_{i \neq j} |v_i| \, |v_j|, \quad \text{by } |\langle \boldsymbol{x}_i, \boldsymbol{x}_j \rangle| \leq \delta$$

$$\leq \delta^k \left( \sum_{i=1}^n |v_i| \right)^2$$

$$\leq n\delta^k, \quad \text{by Cauchy-Schwart's inequlity.}$$

By inverse triangle inequality, we have

$$\|\boldsymbol{K}\boldsymbol{v}\|^2 \geq 1 - n\delta^k.$$

Choose $k_0 \geq \log n / \log(1/\delta)$, then $\lambda_{\min}\{(\boldsymbol{X}^{*k})(\boldsymbol{X}^{*k})^T\} > 0$ for all $k \geq k_0$. $\qquad \square$

### A.5 PROOF OF LEMMA 3.3

*Proof.* Since $\|\boldsymbol{x}_i\| = 1$ and $\boldsymbol{w}_r(0) \sim \mathcal{N}(0, \boldsymbol{I}_d)$, we have $\boldsymbol{x}_i^T \boldsymbol{w}_r(0) \sim N(0, 1)$ for all $i \in [n]$. Let $X_{ir} \triangleq \sigma \left[ \boldsymbol{x}_i^T \boldsymbol{w}_r(0) \right]$ and $Z \sim N(0, 1)$, then for any $|\lambda| \leq 1/\sqrt{2}$, we have

$$\mathbb{E} \exp\{X_{ir}^2 \lambda^2\} = \mathbb{E} \exp\{\sigma \left[ \boldsymbol{x}_i^T \boldsymbol{w}_r(0) \right]^2 \lambda^2\} \leq \mathbb{E} \exp\{Z^2 \lambda^2\} = 1/\sqrt{1 - 2\lambda^2} \leq e^{2t^2},$$

where the first inequality is due tot $|\sigma(x)| \leq |x|$, and the last inequality is by using the numerical inequality $1/(1-x) \leq e^{2x}$. Choose $\lambda \leq \left( \log \sqrt{2} \right)^{1/2}$, we obtain $\mathbb{E}\{X_{ir}^2 \lambda^2\} \leq 2$. By using Markov's inequality, we have for any $t \geq 0$

$$\mathbb{P}\{|X_{ir}| \geq t\} = \mathbb{P}\left\{ |X_{ir}|^2 / \log \sqrt{2} \geq t^2 / \log \sqrt{2} \right\} \leq 2 \exp\left\{ -t^2 \log \sqrt{2} \right\} \leq 2 \exp\left\{ -t^2/4 \right\}.$$

Therefore $X_{ir}$ is a sub-Gaussian random variable with sub-Gaussian norm $\|X_i\|_{\psi_2} \leq 2$ (Vershynin, 2018, Proposition 2.5.2). Then $X_{ir} X_{jr}$ is a sub-exponential random variable with sub-exponential norm $\|X_{ir} X_{jr}\|_{\psi_1} \leq 4$ (Vershynin, 2018, Lemma 2.7.7). Observe that

$$\boldsymbol{G}_{ij}(0) = \boldsymbol{\phi}_i(0)^T \boldsymbol{\phi}_j(0) = \frac{1}{m} \sum_{r=1}^m \sigma \left[ \boldsymbol{x}_i^T \boldsymbol{w}_r(0) \right] \sigma \left[ \boldsymbol{x}_j^T \boldsymbol{w}_r(0) \right] = \frac{1}{m} \sum_{r=1}^m X_{ir} X_{jr}.$$

Since $\boldsymbol{G}_{ij}^{\infty} = \mathbb{E}\left[\boldsymbol{G}_{ij}(0)\right]$, (Vershynin, 2018, Exercise 2.7.10) implies that $\boldsymbol{G}_{ij}(0) - \boldsymbol{G}_{ij}^{\infty}$ is also a zero-mean sub-exponential random variable. It follows from the Bernstein's inequality that

$$
\begin{aligned}
\mathbb{P}\left\{\|\boldsymbol{G}(0) - \boldsymbol{G}^{\infty}\|_2 \geq \frac{\lambda_0}{4}\right\} \leq & \mathbb{P}\left\{\|\boldsymbol{G}(0) - \boldsymbol{G}^{\infty}\|_F \geq \frac{\lambda_0}{4}\right\}\\
=& \mathbb{P}\left\{\|\boldsymbol{G}(0) - \boldsymbol{G}^{\infty}\|_F^2 \geq \left(\frac{\lambda_0}{4}\right)^2\right\}\\
=& \mathbb{P}\left\{\sum_{i,j=1}^{n}\left|\boldsymbol{G}_{ij}(0) - \boldsymbol{G}_{ij}^{\infty}\right|^2 \geq \left(\frac{\lambda_0}{4}\right)^2\right\}\\
\leq& \sum_{i,j=1}^{n}\mathbb{P}\left\{\left|\boldsymbol{G}_{ij}(0) - \boldsymbol{G}_{ij}^{\infty}\right|^2 \geq \left(\frac{\lambda_0}{4n}\right)^2\right\}\\
=& \sum_{i,j=1}^{n}\mathbb{P}\left\{\left|\boldsymbol{G}_{ij}(0) - \boldsymbol{G}_{ij}^{\infty}\right| \geq \frac{\lambda_0}{4n}\right\}\\
\leq& n^2 \cdot 2\exp\left\{-c\lambda_0^2 m/n^2\right\}\\
\leq& \delta,
\end{aligned}
$$

where $c > 0$ is some constant, and we use the facts $\|\boldsymbol{X}\|_2 \leq \|\boldsymbol{X}\|_F$, and $\mathbb{P}\{\sum_{i=1}^{n} x_i \geq \varepsilon\} \leq \sum_{i=1}^{n}\mathbb{P}\{x_i \geq \varepsilon/n\}$. □

## A.6 PROOF OF LEMMA 3.4

*Proof.* By using the 1-Lipschitz continuity of $\sigma(x)$, we have

$$
\begin{aligned}
\|\boldsymbol{G} - \boldsymbol{G}(0)\| =& \frac{1}{m}\|\sigma(\boldsymbol{X}\boldsymbol{W}^T)\sigma(\boldsymbol{X}\boldsymbol{W}^T)^T - \sigma(\boldsymbol{X}\boldsymbol{W}(0)^T)\sigma(\boldsymbol{X}\boldsymbol{W}(0)^T)^T\|\\
\leq& \frac{1}{m}\|\sigma(\boldsymbol{X}\boldsymbol{W}^T)\sigma(\boldsymbol{X}\boldsymbol{W}^T)^T - \sigma(\boldsymbol{X}\boldsymbol{W}^T)\sigma(\boldsymbol{X}\boldsymbol{W}(0)^T)^T\|\\
& + \frac{1}{m}\|\sigma(\boldsymbol{X}\boldsymbol{W}^T)\sigma(\boldsymbol{X}\boldsymbol{W}(0)^T)^T - \sigma(\boldsymbol{X}\boldsymbol{W}(0)^T)\sigma(\boldsymbol{X}\boldsymbol{W}(0)^T)^T\|\\
=& \frac{1}{m}\|\sigma(\boldsymbol{X}\boldsymbol{W}^T)\|\|\sigma(\boldsymbol{X}\boldsymbol{W}^T) - \sigma(\boldsymbol{X}\boldsymbol{W}(0)^T)\|\\
& + \frac{1}{m}\|\sigma(\boldsymbol{X}\boldsymbol{W}^T) - \sigma(\boldsymbol{X}\boldsymbol{W}(0)^T)\|\|\sigma(\boldsymbol{X}\boldsymbol{W}(0)^T)\|\\
\leq& \frac{1}{m}\|\boldsymbol{X}\|\|\boldsymbol{W}\|\|\boldsymbol{X}\|\|\boldsymbol{W} - \boldsymbol{W}(0)\| + \frac{1}{m}\|\boldsymbol{X}\|\|\boldsymbol{W} - \boldsymbol{W}(0)\|\|\boldsymbol{X}\|\|\boldsymbol{W}(0)\|\\
\leq& \frac{4c}{\sqrt{m}}\|\boldsymbol{X}\|^2\|\boldsymbol{W} - \boldsymbol{W}(0)\|\\
\leq& \frac{\lambda_0}{4}.
\end{aligned}
$$

□

## A.7 PROOF OF LEMMA 3.5

*Proof.* It suffices to show the result holds for $\gamma = \min\{\gamma_0, \gamma_0/2c\}$, where $\gamma_0 = 1/2$. Note that Lemma 2.1 still holds if one chooses a larger $c$. Thus, we choose $c \gtrsim \sqrt{\lambda_0}/\|X\|$. We prove by the induction. Suppose that for $0 \leq s \leq t$, the followings hold

(i) $\lambda_{\min}(\boldsymbol{G}(s)) \geq \frac{\lambda_0}{2}$,

(ii) $\|\boldsymbol{u}(s)\| \leq \frac{16c\sqrt{n}}{\lambda_0}\|\hat{\boldsymbol{y}}(0) - \boldsymbol{y}\|$,

(iii) $\|\boldsymbol{v}(s)\| \leq \frac{8c\sqrt{n}}{\lambda_0}\|\hat{\boldsymbol{y}}(0) - \boldsymbol{y}\|$,

(iv) $\|\boldsymbol{W}(s)\| \le 2c\sqrt{m}$,

(v) $\|\boldsymbol{A}(s)\| \le 2c\sqrt{m}$,

(vi) $\|\hat{\boldsymbol{y}}(s) - \boldsymbol{y}\|^2 \le \exp\{-\lambda_0 s\}\|\hat{\boldsymbol{y}}(0) - \boldsymbol{y}\|^2$,

Since $\lambda_{\min}(\boldsymbol{G}(s)) \ge \frac{\lambda_0}{2}$, we have

$$\frac{d}{dt}\|\hat{\boldsymbol{y}}(t) - \boldsymbol{y}\|^2 = -2(\hat{\boldsymbol{y}}(t) - \boldsymbol{y})^T \boldsymbol{H}(t)(\hat{\boldsymbol{y}}(t) - \boldsymbol{y})$$
$$\le -\lambda_0\|\hat{\boldsymbol{y}}(t) - \boldsymbol{y}\|^2$$

Solving the ordinary differential equation yields

$$\|\hat{\boldsymbol{y}}(t) - \boldsymbol{y}\|^2 \le \exp\{-\lambda_0 t\}\|\hat{\boldsymbol{y}}(0) - \boldsymbol{y}\|^2.$$

By using the inductive hypothesis $\|\boldsymbol{W}(s)\| \le 2c\sqrt{m}$, we have

$$\|\boldsymbol{\phi}_i(s)\| = \left\|\frac{1}{\sqrt{m}}\sigma(\boldsymbol{W}(s)\boldsymbol{x}_i)\right\| \le \frac{1}{\sqrt{m}}\|\boldsymbol{W}(s)\|\|\boldsymbol{x}_i\| \le 2c.$$

It follows from Lemma 2.2 with $\gamma_0 = 1/2$ that

$$\|\boldsymbol{z}_i^*(s)\| \le 2\|\boldsymbol{\phi}_i(s)\| \le 4c.$$

Note that

$$\|\nabla_{\boldsymbol{v}}L(s)\| \le \sum_{i=1}^n |\hat{y}_i(s) - y_i|\,\|\boldsymbol{\phi}_i(s)\|$$
$$\le 2c\sum_{i=1}^n |\hat{y}_i(s) - y_i|$$
$$\le 2c\sqrt{n}\|\hat{\boldsymbol{y}}(s) - \boldsymbol{y}\|$$
$$\le 2c\sqrt{n}\exp\{-\lambda_0 s/2\}\|\boldsymbol{y}(0) - \boldsymbol{y}\|$$

and so

$$\|\boldsymbol{v}(t) - \boldsymbol{v}(0)\| \le \int_0^t \|\nabla_{\boldsymbol{v}}L(s)\|ds$$
$$\le 2c\sqrt{n}\|\boldsymbol{y}(0) - \boldsymbol{y}\|\int_0^t \exp\{-\lambda_0 s/2\}ds$$
$$\le \frac{4c\sqrt{n}}{\lambda_0}\|\hat{\boldsymbol{y}}(0) - \boldsymbol{y}\|,$$

Since $\boldsymbol{v}_i(0)$ follows symmetric Bernoulli distribution, then $\|\boldsymbol{v}(0)\| = \sqrt{m}$ and we obtain

$$\|\boldsymbol{v}(t)\| \le \|\boldsymbol{v}(t) - \boldsymbol{v}(0)\| + \|\boldsymbol{v}(0)\| \le \frac{8c\sqrt{n}}{\lambda_0}\|\hat{\boldsymbol{y}}(0) - \boldsymbol{y}\|,$$

where the last inequality is due to $m = \Omega\left(\frac{c^2 n\|\boldsymbol{X}\|^2}{\lambda_0^3}\|\hat{\boldsymbol{y}}(0) - \boldsymbol{y}\|^2\right)$ and $c \gtrsim \sqrt{\lambda_0}/\|\boldsymbol{X}\|$.

Similarly, we have

$$\|\nabla_{\boldsymbol{u}}L(s)\| \le \sum_{i=1}^n |\hat{y}_i(s) - y_i|\,\|\boldsymbol{z}_i^*\|$$
$$\le 4c\sqrt{n}\|\hat{\boldsymbol{y}}(s) - \boldsymbol{y}\|$$
$$\le 4c\sqrt{n}\exp\{-\lambda_0 s/2\}\|\hat{\boldsymbol{y}}(0) - \boldsymbol{y}\|,$$

so that

$$\|\boldsymbol{u}(t) - \boldsymbol{u}(0)\| \leq \int_0^t \|\nabla_{\boldsymbol{u}} L(s)\| ds \leq \frac{8c\sqrt{n}}{\lambda_0} \|\hat{\boldsymbol{y}}(0) - \boldsymbol{y}\|.$$

Since $\boldsymbol{u}_i(0)$ follows symmetric Bernoulli distribution, then $\|\boldsymbol{u}(0)\| = \sqrt{m}$ and we obtain

$$\|\boldsymbol{u}(t)\| \leq \|\boldsymbol{u}(t) - \boldsymbol{u}(0)\| + \|\boldsymbol{u}(0)\| \leq \frac{16c\sqrt{n}}{\lambda_0} \|\hat{\boldsymbol{y}}(0) - \boldsymbol{y}\|.$$

Note that

$$\begin{aligned}
\|\nabla_{\boldsymbol{W}} L(s)\| &\leq \sum_{i=1}^n \frac{1}{\sqrt{m}} |\hat{y}_i(s) - y_i| \, \|\boldsymbol{E}_i(s)\| \left(\|\boldsymbol{U}_i(s)^{-1}\boldsymbol{u}(s)\| + \|\boldsymbol{v}(s)\|\right) \|\boldsymbol{x}_i\| \\
&\leq \frac{64c\sqrt{n}}{\lambda_0} \|\hat{\boldsymbol{y}}(0) - \boldsymbol{y}\| \cdot \sum_{i=1}^n |\hat{y}_i(s) - y_i| \\
&\leq \frac{64cn}{\lambda_0} \|\hat{\boldsymbol{y}}(0) - \boldsymbol{y}\| \cdot \|\hat{\boldsymbol{y}}(s) - \boldsymbol{y}\| \\
&\leq \frac{64cn}{\lambda_0} \|\hat{\boldsymbol{y}}(0) - \boldsymbol{y}\|^2 \cdot \exp\{-\lambda_0 s/2\},
\end{aligned}$$

so that

$$\begin{aligned}
\|\boldsymbol{W}(t) - \boldsymbol{W}(0)\| &\leq \int_0^t \|\nabla_{\boldsymbol{W}} L(s)\| ds \\
&\leq \frac{128cn}{\lambda_0^2 \sqrt{m}} \|\hat{\boldsymbol{y}}(0) - \boldsymbol{y}\|^2 \\
&\leq \frac{\lambda_0 \sqrt{m}}{16c\|\boldsymbol{X}\|^2} \\
&\leq R.
\end{aligned}$$

Therefore, we obtain

$$\|\boldsymbol{W}(t)\| \leq \|\boldsymbol{W}(t) - \boldsymbol{W}(0)\| + \|\boldsymbol{W}(0)\| \leq 2c\sqrt{m},$$

Moreover, it follows from Lemma 3.4 that $\lambda_{\min}\{\boldsymbol{G}(t)\} \geq \frac{\lambda_0}{2}$.

Note that

$$\begin{aligned}
\|\nabla_{\boldsymbol{A}} L(s)\| &\leq \sum_{i=1}^n \frac{\gamma}{\sqrt{m}} |\hat{y}_i(s) - y_i| \, \|\boldsymbol{D}_i\| \|\boldsymbol{U}_i(s)^{-1}\| \|\boldsymbol{u}(s)\| \|\boldsymbol{z}_i^*\| \\
&\leq \frac{32c\sqrt{n}}{\lambda_0\sqrt{m}} \|\hat{\boldsymbol{y}}(0) - \boldsymbol{y}\| \cdot \sum_{i=1}^n |\hat{y}_i(s) - y_i| \\
&\leq \frac{32cn}{\lambda_0\sqrt{m}} \|\hat{\boldsymbol{y}}(0) - \boldsymbol{y}\| \cdot \|\hat{\boldsymbol{y}}(s) - \boldsymbol{y}\| \\
&\leq \frac{32cn}{\lambda_0\sqrt{m}} \|\hat{\boldsymbol{y}}(0) - \boldsymbol{y}\|^2 \cdot \exp\{-\lambda_0 s/2\},
\end{aligned}$$

so that

$$\begin{aligned}
\|\boldsymbol{A}(t) - \boldsymbol{A}(0)\| &\leq \int_0^t \|\nabla_{\boldsymbol{A}} L(s)\| ds \\
&\leq \frac{64cn}{\lambda_0^2 \sqrt{m}} \|\hat{\boldsymbol{y}}(0) - \boldsymbol{y}\|^2.
\end{aligned}$$

Then

$$\|\boldsymbol{A}(t)\| \leq \|\boldsymbol{A}(t) - \boldsymbol{A}(0)\| + \|\boldsymbol{A}(0)\| \leq 2c\sqrt{m}.$$

$\square$

## A.8   PROOF OF LEMMA 3.6

In this section, we prove the result for discrete time analysis or result for gradient descent. Assume $\|\boldsymbol{A}(0)\| \leq c\sqrt{m}$ and $\|\boldsymbol{W}(0)\| \leq c\sqrt{m}$. Further, we assume $\lambda_{\min}(\boldsymbol{G}(0)) \geq \frac{3}{4}\lambda_0$ and we assume $m = \Omega\left(\frac{c^2 n \|\boldsymbol{X}\|^2}{\lambda_0^3}\|\hat{\boldsymbol{y}}(0) - \boldsymbol{y}\|^2\right)$ and choose $0 < \gamma \leq \min\{1/2, 1/4c\}$. Moreover, we assume the stepsize $\alpha = \mathcal{O}\left(\lambda_0/n^2\right)$. We make the inductive hypothesis as follows for all $0 \leq s \leq k$

(i) $\lambda_{\min}(\boldsymbol{G}(s)) \geq \frac{\lambda_0}{2}$,

(ii) $\|\boldsymbol{u}(s)\| \leq \frac{32c\sqrt{n}}{\lambda_0}\|\hat{\boldsymbol{y}}(0) - \boldsymbol{y}\|$,

(iii) $\|\boldsymbol{v}(s)\| \leq \frac{16c\sqrt{n}}{\lambda_0}\|\hat{\boldsymbol{y}}(0) - \boldsymbol{y}\|$,

(iv) $\|\boldsymbol{W}(s)\| \leq 2c\sqrt{m}$,

(v) $\|\boldsymbol{A}(s)\| \leq 2c\sqrt{m}$,

(vi) $\|\hat{\boldsymbol{y}}(s) - \boldsymbol{y}\|^2 \leq (1 - \alpha\lambda_0/2)^s\|\hat{\boldsymbol{y}}(0) - \boldsymbol{y}\|^2$.

*Proof.* Note that Lemma 2.1 still holds if one chooses a larger $c$. Thus, we choose $c \gtrsim \sqrt{\lambda_0}/\|X\|$. By using the inductive hypothesis, we have for any $0 \leq s \leq k$

$$\|\boldsymbol{\phi}_i(s)\| = \|\frac{1}{\sqrt{m}}\sigma(\boldsymbol{W}(s)\boldsymbol{x}_i)\| \leq \frac{1}{\sqrt{m}}\|\boldsymbol{W}(s)\| \leq 2c$$

and

$$\|\boldsymbol{\Phi}(s)\| \leq \|\boldsymbol{\Phi}(s)\|_F = \left(\sum_{i=1}^n \|\boldsymbol{\phi}_i(s)\|^2\right)^{1/2} \leq 2c\sqrt{n}. \tag{33}$$

By using Lemma 2.2, we obtain the upper bound for the equilibrium point $\boldsymbol{z}_i(s)$ for any $0 \leq s \leq k$ as follows

$$\|\boldsymbol{z}_i(s)\| \leq \frac{1}{1 - \gamma_0}\|\boldsymbol{\phi}_i(s)\| = 2\|\boldsymbol{\phi}_i(s)\| \leq 4c,$$

where the last inequality is because we choose $\gamma_0 = 1/2$, and

$$\|\boldsymbol{Z}(s)\| \leq \|\boldsymbol{Z}(s)\|_F = \left(\sum_{i=1}^n \|\boldsymbol{z}_i(s)\|^2\right)^{1/2} = 4c\sqrt{n}. \tag{34}$$

By using the upper bound of $\boldsymbol{\phi}_i(s)$, we obtain for any $0 \leq s \leq k$

$$\begin{aligned}
\|\nabla_{\boldsymbol{v}}L(s)\| &\leq \sum_{i=1}^n |\hat{y}_i(s) - y_i|\,\|\boldsymbol{\phi}_i(s)\| \\
&\leq 2c\sum_{i=1}^n |\hat{y}_i(s) - y_i| \\
&\leq 2c\sqrt{n}\|\hat{\boldsymbol{y}}(s) - \boldsymbol{y}\| \\
&\leq 2c\sqrt{n}(1 - \alpha\lambda_0/2)^{s/2}\|\hat{\boldsymbol{y}}(0) - \boldsymbol{y}\|.
\end{aligned}$$

Let $\beta \triangleq \sqrt{1 - \alpha\lambda_0/2}$. Then the upper bound of $\|\nabla_{\boldsymbol{v}}L(s)\|$ can be written as

$$\|\nabla_{\boldsymbol{v}}L(s)\| \leq 2c\sqrt{n}\beta^s\|\hat{\boldsymbol{y}}(0) - \boldsymbol{y}\|, \tag{35}$$

and

$$\|\boldsymbol{v}(k+1) - \boldsymbol{v}(0)\| \leq \sum_{s=0}^{k} \|\boldsymbol{v}(s+1) - \boldsymbol{v}(s)\| = \alpha \sum_{s=0}^{k} \|\nabla_{\boldsymbol{v}} L(s)\|$$

$$\leq \alpha \cdot 2c\sqrt{n}\|\hat{\boldsymbol{y}}(0) - \boldsymbol{y}\| \cdot \sum_{s=0}^{k} \beta^s$$

$$= \frac{2(1-\beta^2)}{\lambda_0} \cdot 2c\sqrt{n}\|\hat{\boldsymbol{y}}(0) - \boldsymbol{y}\| \frac{1-\beta^{k+1}}{1-\beta}$$

$$\leq \frac{8c\sqrt{n}}{\lambda_0}\|\hat{\boldsymbol{y}}(0) - \boldsymbol{y}\|,$$

where the last inequality we use the facts $\beta < 1$. By triangle inequality, we obtain

$$\|\boldsymbol{v}(k+1)\| \leq \|\boldsymbol{v}(k+1) - \boldsymbol{v}(0)\| + \|\boldsymbol{v}(0)\| \leq \frac{16c\sqrt{n}}{\lambda_0}\|\hat{\boldsymbol{y}}(0) - \boldsymbol{y}\|,$$

which proves the result (iii). Similarly, we can upper bound the gradient of $\boldsymbol{u}$

$$\|\nabla_{\boldsymbol{u}} L(s)\| \leq \sum_{i=1}^{n} |\hat{y}_i(s) - y_i| \, \|\boldsymbol{z}_i\| \leq 4c\sqrt{n}\|\hat{\boldsymbol{y}}(s) - \boldsymbol{y}\| \leq 4c\sqrt{n}\beta^s\|\hat{\boldsymbol{y}}(0) - \boldsymbol{y}\| \qquad (36)$$

so that

$$\|\boldsymbol{u}(k+1) - \boldsymbol{u}(0)\| \leq \frac{16c\sqrt{n}}{\lambda_0}\|\hat{\boldsymbol{y}} - \boldsymbol{y}\|,$$

and

$$\|\boldsymbol{u}(k)\| \leq \|\boldsymbol{u}(k) - \boldsymbol{u}(0)\| + \|\boldsymbol{u}(0)\| \leq \frac{32c\sqrt{n}}{\lambda_0}\|\hat{\boldsymbol{y}}(0) - \boldsymbol{y}\|.$$

The result (ii) is also obtained.

By using the inductive hypothesis, we can upper bound the gradient of $\boldsymbol{W}$ as follows

$$\|\nabla_{\boldsymbol{W}} L(s)\| \leq \sum_{i=1}^{n} \frac{1}{\sqrt{m}} |\hat{y}_i(s) - y_i| \, \|\boldsymbol{E}_i(s)\| \left( \|\boldsymbol{U}_i(s)^{-1}\boldsymbol{u}(s)\| + \|\boldsymbol{v}(s)\| \right) \|\boldsymbol{x}_i\|$$

$$\leq \frac{128c\sqrt{n}}{\lambda_0\sqrt{m}}\|\hat{\boldsymbol{y}}(0) - \boldsymbol{y}\| \sum_{i=1}^{n} |\hat{y}_i(s) - y_i|$$

$$\leq \frac{128cn}{\lambda_0\sqrt{m}}\|\hat{\boldsymbol{y}}(0) - \boldsymbol{y}\| \cdot \|\hat{\boldsymbol{y}}(s) - \boldsymbol{y}\|$$

$$\leq \frac{128cn}{\lambda_0\sqrt{m}}\|\hat{\boldsymbol{y}}(0) - \boldsymbol{y}\|^2 \cdot \beta^s, \qquad (37)$$

so that

$$\|\boldsymbol{W}(k+1) - \boldsymbol{W}(0)\| \leq \alpha \sum_{s=0}^{k} \|\nabla_{\boldsymbol{W}} L(s)\|$$

$$\leq \alpha \cdot \frac{128cn}{\lambda_0^2\sqrt{m}}\|\hat{\boldsymbol{y}}(0) - \boldsymbol{y}\|^2 \cdot \sum_{s=0}^{k} \beta^s$$

$$\leq \frac{512cn}{\lambda_0^2\sqrt{m}}\|\hat{\boldsymbol{y}}(0) - \boldsymbol{y}\|^2$$

$$\leq \frac{\sqrt{m}\lambda_0}{16c\|\boldsymbol{X}\|^2}$$

$$\leq R,$$

where the third inequality holds is because $m$ is large, *i.e.*, $m = \Theta\left(\frac{c^2 n \|\boldsymbol{X}\|^2}{\lambda_0^3} \|\hat{\boldsymbol{y}}(0) - \boldsymbol{y}\|^2\right)$. To simplify the notation, we assume

$$m = \frac{C c^2 n \|\boldsymbol{X}\|^2}{\lambda_0^3} \|\hat{\boldsymbol{y}}(0) - \boldsymbol{y}\|^2 \tag{38}$$

for some large number $C > 0$. Moreover, we obtain

$$\|\boldsymbol{W}(k+1)\| \leq \|\boldsymbol{W}(k+1) - \boldsymbol{W}(0)\| + \|\boldsymbol{W}(0)\| \leq 2c\sqrt{m},$$

Therefore, it follows from Lemma 3.4 that $\lambda_{\min}\{\boldsymbol{G}(k+1)\} \geq \frac{\lambda_0}{2}$. Thus, the results (i) and (iv) are established.

By using similar argument, we can upper bound the gradient of $\boldsymbol{A}$ as follows Note that

$$
\begin{aligned}
\|\nabla_{\boldsymbol{A}} L(s)\| &\leq \sum_{i=1}^{n} \frac{\gamma}{\sqrt{m}} |\hat{y}_i(s) - y_i| \|\boldsymbol{D}_i\| \|\boldsymbol{U}_i(s)^{-1}\| \|\boldsymbol{u}(s)\| \|\boldsymbol{z}_i^*\| \\
&\leq \frac{64 c \sqrt{n}}{\lambda_0 \sqrt{m}} \|\hat{\boldsymbol{y}}(0) - \boldsymbol{y}\| \cdot \sum_{i=1}^{n} |\hat{y}_i(s) - y_i| \\
&\leq \frac{64 c n}{\lambda_0 \sqrt{m}} \|\hat{\boldsymbol{y}}(0) - \boldsymbol{y}\| \cdot \|\hat{\boldsymbol{y}}(s) - \boldsymbol{y}\| \\
&\leq \frac{64 c n}{\lambda_0 \sqrt{m}} \|\hat{\boldsymbol{y}}(0) - \boldsymbol{y}\|^2 \cdot \beta^s,
\end{aligned}
$$

so that

$$
\begin{aligned}
\|\boldsymbol{A}(k+1) - \boldsymbol{A}(0)\| &\leq \alpha \sum_{s=0}^{k} \|\nabla_{\boldsymbol{A}} L(s)\| \\
&\leq \alpha \cdot \frac{64 c n}{\lambda_0 \sqrt{m}} \|\hat{\boldsymbol{y}}(0) - \boldsymbol{y}\|^2 \cdot \sum_{s=0}^{k} \beta^s \\
&\leq \frac{256 c n}{\lambda_0^2 \sqrt{m}} \|\hat{\boldsymbol{y}}(0) - \boldsymbol{y}\|^2.
\end{aligned}
$$

Since $m = \frac{C c^2 n \|\boldsymbol{X}\|^2}{\lambda_0^3} \|\hat{\boldsymbol{y}}(0) - \boldsymbol{y}\|^2$ and $c, C > 0$ are large enough, we have

$$\|\boldsymbol{A}(k+1)\| \leq \|\boldsymbol{A}(k+1) - \boldsymbol{A}(0)\| + \|\boldsymbol{A}(0)\| \leq 2c\sqrt{m}.$$

Therefore, the result (v) is obtained and the equilibrium points $\boldsymbol{z}_i(k+1)$ exists for all $i \in [n]$.

To establish the result (vi), we need to derive the bounds between equilibrium points $\boldsymbol{Z}(k)$ and feature vectors $\boldsymbol{\Phi}(k)$. We firstly bound the difference between equilibrium points $\boldsymbol{z}_i(k+1)$ and $\boldsymbol{z}_i(k)$. For any $\ell \geq 1$, we have

$$
\begin{aligned}
\|\boldsymbol{z}_i^{\ell+1}(k+1) - \boldsymbol{z}_i^{\ell+1}(k)\| &= \|\sigma\left[\tilde{\gamma}\boldsymbol{A}(k+1)\boldsymbol{z}_i^{\ell}(k+1) + \phi_i(k+1)\right] - \sigma\left[\tilde{\gamma}\boldsymbol{A}(k)\boldsymbol{z}_i^{\ell}(k) + \phi_i(k)\right]\| \\
&\leq \|\tilde{\gamma}\boldsymbol{A}(k+1)\boldsymbol{z}_i^{\ell}(k+1) + \phi_i(k+1) - \tilde{\gamma}\boldsymbol{A}(k)\boldsymbol{z}_i^{\ell}(k) - \phi_i(k)\| \\
&\leq \tilde{\gamma}\|\boldsymbol{A}(k+1)\boldsymbol{z}_i^{\ell}(k+1) - \boldsymbol{A}(k)\boldsymbol{z}_i^{\ell}(k)\| + \|\phi_i(k+1) - \phi_i(k)\|,
\end{aligned}
$$

where the first term can be bounded as follows

$$
\begin{aligned}
&\tilde{\gamma}\|\boldsymbol{A}(k+1)\boldsymbol{z}_i^{\ell}(k+1) - \boldsymbol{A}(k)\boldsymbol{z}_i^{\ell}(k)\| \\
&\leq \tilde{\gamma}\|\boldsymbol{A}(k+1) - \boldsymbol{A}(k)\| \|\boldsymbol{z}_i^{\ell}(k+1)\| + \tilde{\gamma}\|\boldsymbol{A}(k)\| \|\boldsymbol{z}_i^{\ell}(k+1) - \boldsymbol{z}_i^{\ell}(k)\| \\
&\leq \tilde{\gamma}\alpha\|\nabla_{\boldsymbol{A}} L(k)\|(4c) + \tilde{\gamma}\|\boldsymbol{A}(k)\| \|\boldsymbol{z}_i^{\ell}(k+1) - \boldsymbol{z}_i^{\ell}(k)\| \\
&\leq \frac{64 \alpha c n}{\lambda_0 m} \|\hat{\boldsymbol{y}}(0) - \boldsymbol{y}\|^2 \beta^k + (1/2)\|\boldsymbol{z}_i^{\ell}(k+1) - \boldsymbol{z}_i^{\ell}(k)\|,
\end{aligned}
$$

and the second term is bounded as follows

$$\|\boldsymbol{\phi}_i(k+1) - \boldsymbol{\phi}_i(k)\|$$
$$= \frac{1}{\sqrt{m}} \|\sigma[\boldsymbol{W}(k+1)\boldsymbol{x}_i] - \sigma[\boldsymbol{W}(k)\boldsymbol{x}_i]\|$$
$$\leq \frac{1}{\sqrt{m}} \|\boldsymbol{W}(k+1) - \boldsymbol{W}(k)\| \|\boldsymbol{x}_i\|$$
$$\leq \frac{\alpha}{\sqrt{m}} \|\nabla_{\boldsymbol{W}} L(k)\|$$
$$\leq \frac{128\alpha cn}{\lambda_0 m} \|\hat{\boldsymbol{y}}(0) - \boldsymbol{y}\|^2 \cdot \beta^k.$$

Thus, we obtain

$$\|\boldsymbol{z}_i^{\ell+1}(k+1) - \boldsymbol{z}_i^{\ell+1}(k)\| \leq (1/2)\|\boldsymbol{z}_i^{\ell}(k+1) - \boldsymbol{z}_i^{\ell}(k)\| + \frac{256\alpha cn}{\lambda_0 m} \|\hat{\boldsymbol{y}}(0) - \boldsymbol{y}\|^2 \cdot \beta^k$$
$$\leq (1/2)^{\ell} \|\boldsymbol{z}_i^1(k+1) - \boldsymbol{z}_i^1(k)\| + \frac{256\alpha cn}{\lambda_0 m} \|\hat{\boldsymbol{y}}(0) - \boldsymbol{y}\|^2 \cdot \beta^k \cdot \sum_{j=0}^{\infty} 2^{-j}$$
$$\leq (1/2)^{\ell} \|\boldsymbol{z}_i^1(k+1) - \boldsymbol{z}_i^1(k)\| + \frac{512\alpha cn}{\lambda_0 m} \|\hat{\boldsymbol{y}}(0) - \boldsymbol{y}\|^2 \cdot \beta^k.$$

By letting $\ell \to \infty$, we obtain

$$\|\boldsymbol{z}_i(k+1) - \boldsymbol{z}_i(k)\| \leq \frac{512\alpha cn}{\lambda_0 m} \|\hat{\boldsymbol{y}}(0) - \boldsymbol{y}\|^2 \cdot \beta^k.$$

By using the Cauchy-Schwartz's inequality, we have

$$\|\boldsymbol{Z}(k+1) - \boldsymbol{Z}(k)\| \leq \|\boldsymbol{Z}(k+1) - \boldsymbol{Z}(k)\|_F \leq \frac{512\alpha cn^{3/2}}{\lambda_0 m} \|\hat{\boldsymbol{y}}(0) - \boldsymbol{y}\|^2 \cdot \beta^k. \tag{39}$$

In addition, we will also bound the difference in $\boldsymbol{\phi}_i(k+1)$ and $\boldsymbol{\phi}_i(k)$. Note that

$$\|\boldsymbol{\phi}_i(k+1) - \boldsymbol{\phi}_i(k)\| = \frac{1}{\sqrt{m}} \|\sigma[\boldsymbol{W}(k+1)\boldsymbol{x}_i] - \sigma[\boldsymbol{W}(k)\boldsymbol{x}_i]\| \leq \frac{128\alpha cn}{\lambda_0 m} \|\hat{\boldsymbol{y}}(0) - \boldsymbol{y}\|^2 \cdot \beta^k,$$

so that

$$\|\boldsymbol{\Phi}(k+1) - \boldsymbol{\Phi}(k)\| \leq \|\boldsymbol{\Phi}(k+1) - \boldsymbol{\Phi}(k)\|_F \leq \frac{128\alpha cn^{3/2}}{\lambda_0 m} \|\hat{\boldsymbol{y}}(0) - \boldsymbol{y}\|^2 \cdot \beta^k \tag{40}$$

Now, we are ready to establish the result (vi). Note that

$$\|\hat{\boldsymbol{y}}(k+1) - \boldsymbol{y}\|^2 = \|\hat{\boldsymbol{y}}(k+1) - \hat{\boldsymbol{y}}(k) + \hat{\boldsymbol{y}}(k) - \boldsymbol{y}\|^2$$
$$= \|\hat{\boldsymbol{y}}(k+1) - \hat{\boldsymbol{y}}(k)\|^2 + 2\langle \hat{\boldsymbol{y}}(k+1) - \hat{\boldsymbol{y}}(k), \hat{\boldsymbol{y}}(k) - \boldsymbol{y}\rangle + \|\hat{\boldsymbol{y}}(k) - \boldsymbol{y}\|^2.$$

In the rest of this proof, we will bound each term in the above inequality. By the prediction rule of $\hat{\boldsymbol{y}}$, we can bound the difference between $\hat{\boldsymbol{y}}(k+1)$ and $\hat{\boldsymbol{y}}(k)$ as follows

$$\|\hat{\boldsymbol{y}}(k+1) - \hat{\boldsymbol{y}}(k)\| = \|\boldsymbol{Z}(k+1)\boldsymbol{u}(k+1) + \boldsymbol{\Phi}(k+1)\boldsymbol{v}(k+1) - \boldsymbol{Z}(k)\boldsymbol{u}(k) - \boldsymbol{\Phi}(k)\boldsymbol{v}(k)\|$$
$$\leq \|\boldsymbol{Z}(k+1)\boldsymbol{u}(k+1) - \boldsymbol{Z}(k)\boldsymbol{u}(k)\| + \|\boldsymbol{\Phi}(k+1)\boldsymbol{v}(k+1) - \boldsymbol{\Phi}(k)\boldsymbol{v}(k)\|,$$

where the first term can be bounded as follows by using (34), 36, 38, 39, hypothesis (ii), and a large constant $C_0 > 0$

$$\|\boldsymbol{Z}(k+1)\| \|\boldsymbol{u}(k+1) - \boldsymbol{u}(k)\| + \|\boldsymbol{Z}(k+1) - \boldsymbol{Z}(k)\| \|\boldsymbol{u}(k)\|$$
$$= \alpha \|\boldsymbol{Z}(k+1)\| \|\nabla_{\boldsymbol{u}} L(k)\| + \|\boldsymbol{Z}(k+1) - \boldsymbol{Z}(k)\| \|\boldsymbol{u}(k)\|$$
$$\leq \alpha C_0 c^2 n \|\hat{\boldsymbol{y}}(0) - \boldsymbol{y}\| \cdot \beta^k,$$

and the second term is bounded as follows by using (33), 35, 40, 38, hypothesis (iii), and a large constant $C_0 > 0$

$$\|\boldsymbol{\Phi}(k+1)\|\|\boldsymbol{v}(k+1) - \boldsymbol{v}(k)\| + \|\boldsymbol{\Phi}(k+1) - \boldsymbol{\Phi}(k)\|\|\boldsymbol{v}(k)\|$$
$$=\alpha\|\boldsymbol{\Phi}(k+1)\|\|\nabla_{\boldsymbol{v}}L(k)\| + \|\boldsymbol{\Phi}(k+1) - \boldsymbol{\Phi}(k)\|\|\boldsymbol{v}(k)\|$$
$$\leq\alpha C_0 c^2 n\|\hat{\boldsymbol{y}}(0) - \boldsymbol{y}\| \cdot \beta^k.$$

Therefore, we have

$$\|\hat{\boldsymbol{y}}(k+1) - \hat{\boldsymbol{y}}(k)\| \leq \alpha C_0 c^2 n\|\hat{\boldsymbol{y}}(0) - \boldsymbol{y}\| \cdot \beta^k, \tag{41}$$

where the scalar 2 is absorbed in $C_0$ and the constant $C_0$ is difference from $C$.

Let $\boldsymbol{g} \triangleq \boldsymbol{Z}(k)\boldsymbol{u}(k+1) + \boldsymbol{\Phi}(k)\boldsymbol{v}(k+1)$. Then we have

$$\langle\hat{\boldsymbol{y}}(k+1) - \hat{\boldsymbol{y}}(k), \hat{\boldsymbol{y}}(k) - \boldsymbol{y}\rangle = \langle\hat{\boldsymbol{y}}(k+1) - \boldsymbol{g}, \hat{\boldsymbol{y}}(k) - \boldsymbol{y}\rangle + \langle\boldsymbol{g} - \hat{\boldsymbol{y}}(k), \hat{\boldsymbol{y}}(k) - \boldsymbol{y}\rangle.$$

Let us bound each term individually. By using the Cauchy-Schwartz inequality, we have

$$\langle\hat{\boldsymbol{y}}(k+1) - \boldsymbol{g}, \hat{\boldsymbol{y}}(k) - \boldsymbol{y}\rangle$$
$$= \langle(\boldsymbol{Z}(k+1) - \boldsymbol{Z}(k))\boldsymbol{u}(k+1), \hat{\boldsymbol{y}}(k) - \boldsymbol{y}\rangle + \langle(\boldsymbol{\Phi}(k+1) - \boldsymbol{\Phi}(k))\boldsymbol{v}(k+1), \hat{\boldsymbol{y}}(k) - \boldsymbol{y}\rangle$$
$$\leq (\|\boldsymbol{Z}(k+1) - \boldsymbol{Z}(k)\|\|\boldsymbol{u}(k+1)\| + \|\boldsymbol{\Phi}(k+1) - \boldsymbol{\Phi}(k)\|\|\boldsymbol{v}(k+1)\|) \|\hat{\boldsymbol{y}}(k) - \boldsymbol{y}\|$$
$$\leq\alpha C_0 c^2 n\|\hat{\boldsymbol{y}}(0) - \boldsymbol{y}\| \cdot \beta^k\|\hat{\boldsymbol{y}}(k) - \boldsymbol{y}\|, \quad \text{by (34), 36, 38, 39}$$
$$\leq\alpha C_0 c^2 n \cdot \beta^{2k}\|\hat{\boldsymbol{y}}(0) - \boldsymbol{y}\|^2. \tag{42}$$

By using $\nabla_{\boldsymbol{u}}L(k) = \boldsymbol{Z}(k)^T(\hat{\boldsymbol{y}}(k) - \boldsymbol{y})$, $\nabla_{\boldsymbol{v}}L(k) = \boldsymbol{\Phi}(k)^T(\hat{\boldsymbol{y}}(k) - \boldsymbol{y})$ and $\lambda_{\min}(\boldsymbol{G}(k)) \geq \lambda_0/2$, we get

$$\langle\boldsymbol{g} - \hat{\boldsymbol{y}}(k), \hat{\boldsymbol{y}}(k) - \boldsymbol{y}\rangle = -\alpha(\hat{\boldsymbol{y}}(k) - \boldsymbol{y})^T \left[\boldsymbol{Z}(k)\boldsymbol{Z}(k)^T + \boldsymbol{\Phi}(k)\boldsymbol{\Phi}(k)^T\right] (\hat{\boldsymbol{y}}(k) - \boldsymbol{y})$$
$$\leq -\frac{\alpha\lambda_0}{2}\|\hat{\boldsymbol{y}}(k) - \boldsymbol{y}\|^2. \tag{43}$$

By combining the inequalities (41), 42, 43, we obtain

$$\|\hat{\boldsymbol{y}}(k+1) - \boldsymbol{y}\|^2 \leq \left(1 - \alpha\left[\lambda_0 - C_0 c^2 n - \alpha C_0^2 c^4 n^2\right]\right)\beta^{2k}\|\hat{\boldsymbol{y}}(0) - \boldsymbol{y}\|^2$$
$$\leq \left(1 - \frac{\alpha\lambda_0}{2}\right) \cdot \beta^{2k}\|\hat{\boldsymbol{y}}(0) - \boldsymbol{y}\|^2$$
$$= \left(1 - \frac{\alpha\lambda_0}{2}\right)^{k+1} \|\hat{\boldsymbol{y}}(0) - \boldsymbol{y}\|^2,$$

where the second inequality is by $\alpha = \mathcal{O}\left(\frac{\lambda_0}{n^2}\right)$. This proves the result (vi) and complete the proof.

$\square$

