# OpenReview forum: "A global convergence theory for deep ReLU implicit networks via over-parameterization"
_ICLR.cc/2022/Conference — ICLR 2022 Poster_

### Official Review · Reviewer_pFXH · 2021-11-01

**Correctness:** 4
**Technical Novelty And Significance:** 3
**Empirical Novelty And Significance:** Not applicable
**Recommendation:** 8
**Confidence:** 3

**Main Review:**


Pros.
1.  This paper makes a clear contribution to proving the convergence of gradient descent for an implicit neural network with ReLU activation with infinite layers and finite width. I think that using implicit differentiation for the partial gradient at the fixed point is interesting, enabling the proof of convergence by showing the strictly positive-definite of the Gram matrix $\boldsymbol{G}(t)$.

2.  To ensure the strictly positive-definite of the Gram matrix $\boldsymbol{G}(t)$,   the required number  $m= \tilde \Omega(n^2)$ is competitive or better than recent results for the finite-layered neural network.  In addition,  the result in this paper hold for infinite layers.

 3. The paper is well organized and clearly written.

Cons.

1. The  gradient $\nabla _\boldsymbol{A}L $ and $\nabla _\boldsymbol{u}L $  involves  equilibrium point $\boldsymbol{z}$.  However, it is not easy to achieve the equilibrium point explicitly.  How to compute the gradient for training? Does it need an approximation or a solver for the equilibrium point? It seems to me that a solver demands a high time cost.  Does it scale to a large-scale problem?  It is interesting to discuss the relationship (advantage/disadvantage) compared with neural networks with explicit proximal mapping architecture.  e.g.,  (Lyu et al. 2021)  has a similar NN architecture

$ \boldsymbol{y}_{t+1} = h( \boldsymbol{D}^\top_t\boldsymbol{x} +  (\boldsymbol{I}-\boldsymbol{D}^\top_t\boldsymbol{D}_t)\boldsymbol{y}_t   )$
with

$\boldsymbol{y}_{0} = \boldsymbol{0}$

When sharing weight $ \boldsymbol{D}_t=  \boldsymbol{D}$, and set $ \tilde \gamma  \boldsymbol{A} = \boldsymbol{I}-\boldsymbol{D}^\top \boldsymbol{D} $, it seems to be  a finite-step updated  NN   instead of the fixed point $\boldsymbol{z}^*$ in  Eq.(3).

2. ReLU function $f(x) = max(0,x)$ is not differentiable at point $x=0$. How does this influence the continuous time ODE analysis for the linear convergence?

Minor

Typos.   In the proof of Lemma 2.2 in Appendix A.1,  it should be $\sigma (\tilde \gamma \boldsymbol{A} \boldsymbol{z}^{l-1} + \phi  )$ instead of $\sigma (\tilde \gamma \boldsymbol{A} \boldsymbol{z}^{l-1} - \phi  )$.

Lyu et al.  Neural Optimization Kernel Towards Robust Deep Learning.

**Summary Of The Paper:**


In this paper, the authors theoretically analyze the convergence of gradient descent for an implicit neural network with infinite layers with ReLU activation.  The authors show the unique fixed point of the infinite-layered mapping when the weight matrix $\boldsymbol{A}$ has a properly bounded spectral norm.  Using implicit differentiation, the authors show the partial gradient at the fixed point.  Furthermore, the authors show the linear convergence rate by proving the strictly positive-definite of the Gram matrix $\boldsymbol{G}(t)$  (and $\boldsymbol{H}(t)$).

**Summary Of The Review:**


Overall,  I think this paper makes a clear contribution to proving the convergence of gradient descent for an implicit neural network with ReLU activation with infinite layers.  So I recommend acceptance.

---

> ### Author Response · Authors · 2021-11-19
> **Response to Reviewer pFXH**
>
> Thank you for the positive feedback and comments that our paper provides "competitive or better than recent results for the finite-layered neural network" and "the result in this paper hold for infinite layers''. Also, we highly appreciate the reviewer’s comments on our weaknesses and related works, as they allow us to improve the quality of the paper. We revised the paper according to your suggestions and our point-to-point responses are as follows:
>
> > **Your concern**: 1. The gradient $\nabla_{A}L$ and $\nabla_{u}L$
> involves equilibrium point $z$. However, it is not easy to achieve the equilibrium point explicitly. How to compute the gradient for training? Does it need an approximation or a solver for the equilibrium point? It seems to me that a solver demands a high time cost. Does it scale to a large-scale problem?
>
> **Our response**: In general, the equilibrium point can be found in two ways. One way is to use fixed-point iteration. One can create a sequence $\{z^{\ell}\}$ by $z^{\ell+1}=F(z^{\ell};A,W)$, where $F$ denotes the transition defined in Eq. (1). As discussed in Lemma 2.2, $F$ is a contraction, so that $\{z^{\ell}\}$ converges to the unique equilibrium point $z$ linearly. Since $A$ is appropriately scaled, the convergence rate only depends on the $\gamma$-value we choose. Thus, the fixed-point iteration can scale to large-scale problems. Since the equilibrium point $z$ is a root of the equation $f(z,A,W):=z-F(z;A,W)$, one can also use a numerical root-finding method (for example, Ref. [1-2]) to find the equilibrium point (possibly up to some numerical error). The scalability of this method depends on the root-finding method itself.
>
> There are usually two ways to compute the gradient in practice. Analogous to explicit DNN, we can compute and store all the intermediate values and apply backpropagation to compute the gradient. However, this method may need a lot of memory to store the intermediate values. Instead, we can compute the gradients directly by using Eq. (10-13). Note that we do not need the inverse explicitly to compute the gradients. Instead, the gradients can be computed by solving another equilibrium equation with vector-Jacobian product operations. The tutorial [3] provides detailed descriptions, codes, slides, videos about how to compute the equilibrium points and the gradients. Moreover, a Quasi-Newton-like method is introduced in [1] to compute the equilibrium points and gradients.
> Surprisingly, they observed that the backward pass to compute a gradient is much more efficient, while the forward pass takes an increasingly longer time as the number of training epochs increases. The partial reason is that the authors in [1] may not scale the weight matrix $A$ appropriately, since they only initialize $A_{ij}$ from the normal distribution $N(0,0.05)$, and they use 24 million parameters in total.
>
> > **Your comment**: 2. It is interesting to discuss the relationship (advantage/disadvantage) compared with neural networks with explicit proximal mapping architecture. e.g., (Lyu et al. 2021) has a similar NN architecture
>
> **Our response**: The contribution of our work is orthogonal to the results from (Lyu et al. 2021). The main contribution of (Lyu et al. 2021) is that they analyze the depth benefit of a neural network by connecting a network with a functional optimization problem. Their results mainly focus on the expressive power of the neural network. Our work focuses on the convergence behavior of training an implicit neural network by gradient descent.
> On the other hand, it would be interesting to apply the techniques and knowledge from one paper to the other, since the proximal gradient descent in (Lyu et al. 2021) can be considered as an implicit layer by forcing $T\rightarrow \infty$. We will explore this point in the future.
>
> > **Your comment**: 3. ReLU function is not differentiable at point $x=0$. How does this influence the continuous time ODE analysis for the linear convergence?''
>
> **Our response**: ReLU activation is differentiable everywhere except at $x=0$. The common strategy used in the community of deep learning with ReLU activation is to compute subgradient of ReLU, i.e., $\sigma^{\prime}(x)=\mathrm{1}_{x\geq0\}$ [4-8]. By using the notion of subgradient, the diagonal matrices $D_i$ and $E_i$ are well-defined. Therefore, the ODE of prediction $\hat{y}$ is also well-defined. As long as the Gram matrix $H(t)$ is strictly positive definite over all iterations, the (locally) Polyak-Łojasiewicz inequality is established, then Ref. [9] implies the gradient convergence is obtained.
>
> > **Your comment**: 4. Typos. In the proof of Lemma 2.2 in Appendix A.1, it should be $\sigma(\tilde{\gamma}Az^{\ell-1}+\phi)$ instead of $\sigma(\tilde{\gamma}Az^{\ell-1}-\phi)$.
>
> **Our response**: Yes, that is a typo. We have corrected this typo in our revision. We thank the reviewer for catching the typos, which allows us to improve the quality of the paper.

---

> > ### Author Response · Authors · 2021-11-19
> > **Response to Reviewer pFXH (continued)**
> >
> > [1] Deep Equilibrium Models, NeurIPS 2019
> >
> > [2] Multiscale Deep Equilibrium Models, NeurIPS 2020
> >
> > [3] http://implicit-layers-tutorial.org/
> >
> > [4] A Convergence Theory for Deep Learning via Over-Parameterization, ICML 2019
> >
> > [5] Gradient Descent Provably Optimizes Over-parameterized Neural Networks, ICLR 2018
> >
> > [6] An Improved Analysis of Training Over-parameterized Deep Neural Networks, NeurIPS 2019
> >
> > [7] On the Proof of Global Convergence of Gradient Descent for Deep ReLU Networks with Linear Widths, ICML 2021
> >
> > [8] Gradient descent finds global minima of deep neural networks, ICML 2019
> >
> > [9] Linear Convergence of Gradient and Proximal-Gradient Methods Under the Polyak-Łojasiewicz Condition, ECML 2016
> >
> > [10] Neural Optimization Kernel: Towards Robust Deep Learning, arXiv 2021

---

### Official Review · Reviewer_tTX4 · 2021-11-01

**Correctness:** 4
**Technical Novelty And Significance:** 3
**Empirical Novelty And Significance:** 3
**Recommendation:** 8
**Confidence:** 3

**Main Review:**

[Strength]
- The paper studies the very important problem of convergence of training for implicit models. The problem is non-trivial even given recent advances in relavent proofs for explicit forward-feeding because of the well-posedness issue in implicit models which presents because implicit models can be seen as infinitely deep neural networks.
- The authors show that by puting a proper simple scaling factor to the weights, the well-posedness property can be maintained throughout the training process with no extra regularization or projection steps. This enables the proof of training convergence for implicit models.
- Thorough mathematical proofs for both the continuous setting and the practical discrete setting are given in the paper to support the results which are then varified by numerical experiments.

[Weekness]
- There is a typo in the notations section. I suppose it is lambda_max(A) <= ||A|| since A is not assumed to be positive semidefinite?

**Summary Of The Paper:**

The paper presents a proof of exponential convergence to global optimality in the over-parametrization settings for an implicit model with scaled weights parameters. Although existing work has established similar proofs for feedforward explicit neural networks, such methods don't work with non-linearly activated implicit models where the well-posedness issue poses challenges to the training process. The authors shows that by scaling the weights, well-posedness can be ensured. The convergence result is obtained first on continuous settings and is then extended to discrete settings. Numerical experiments on real datasets confirms the finding.

**Summary Of The Review:**

The paper sets the foundation for the training theories for implicit models. Though some common techniques are employed to in the derivations, the authors successfully tackle the key issue of well-posedness to make the convergence result possible. The reviewer believes this result is significant for implicit models which have become increasingly popular in the community.

---

> ### Author Response · Authors · 2021-11-19
> **Response to Reviewer tTX4**
>
> Thank you for the positive feedback and remarks on the importance and the challenges of theoretically understanding implicit models by commenting that "The paper studies the very important problem of convergence of training for implicit models", "The paper sets the foundation for the training theories for implicit models", "the well-posedness issue poses challenges to the training process", "implicit models can be seen as infinitely deep neural networks". Also, we highly appreciate the reviewer’s comments on our weakness, as they allow us to improve the paper. Thank you! We have revised the paper according to all of your suggestions as described below.
>
> >**Your concern**: 1. There is a typo in the notations section. I suppose it is $\lambda_{\max}(A) \leq ||A||$ since $A$ is not assumed to be positive semidefinite?
>
> **Our response**: Yes, it is a typo. Here, $A$ is a square matrix, and $\lambda_{\max}(A)\leq ||A||$. We would like to thank the reviewer to point out the typos and mistakes in the paper as it helps us to improve the quality of the paper.

---

### Official Review · Reviewer_XwkY · 2021-11-04

**Correctness:** 4
**Technical Novelty And Significance:** 3
**Empirical Novelty And Significance:** Not applicable
**Recommendation:** 6
**Confidence:** 2

**Main Review:**

To be honest I am not familiar with both theorems and applications of implicit networks. Seems that it is empirically successful but lacks theoretical understanding, then I think this paper provides a good starting point. Under the form (1) and (2), the paper first shows the existence of the equilibrium point given $\|A\|$ is bounded. Then the proof of the convergence is similar to DNN:

1. Write down the dynamics (15) and show that one of the terms in $H$ has lower-bounded eigenvalues. (The following calculation heavily relies on the form (2) of $\phi$. Is this commonly used in applications?)
2. For sufficiently large $m$ (over-parameterization), the random initialization $G(0)$ is close to the infinite-wide $G^\infty$.
3. The lower bound of $G(t)$ gives the linear convergence rate, then the fast convergence indeed guarantees that $G(t)$ is not far from $G(0)$ during the trajectory.

Despite that the approach is sort of standard (and the dynamics seems to be simpler than DNN since all the layers share the same weights?), the proof is not trivial and the theorem is good as it gives the first theoretical optimization result for implicit networks. The paper also implements numerical experiments on several standard image dataset to show the effectiveness of the implicit networks.

**PS:** Thank the authors for the detailed response!

**Summary Of The Paper:**

This paper theoretically analyzes the optimization of deep ReLU implicit networks. It first shows the well-posedness of the problem, i.e., the existence and uniqueness of the equilibrium point, then proves that under over-parameterization, both continuous and discrete GD have global convergence in a linear rate, and the approach is similar to the standard proof for DNN.

**Summary Of The Review:**

The paper proves the convergence of the optimizing nonlinear implicit networks. The proof techniques follow the standard approach for DNN, and I think it is a good starting point for the theoretical analysis of implicit networks.

---

> ### Author Response · Authors · 2021-11-19
> **Response to Reviewer XwkY**
>
> Thank you for the feedback and positive comments that our paper provides “a good starting point for the theoretical analysis of implicit networks” and “the proof is not trivial and the theorem is good as it gives the first theoretical optimization result for implicit networks”. Below we respond to your comments and questions by the order in which they appear.
>
> >**Your comment**: 1. The following calculation heavily relies on the form (2) of $\phi$. Is this commonly used in applications?
>
> **Our response**: The form $(2)$ of $\phi$ is a very simple nonlinear feature map to map the input $x$ to the desired feature vector $\phi:=\phi(x)$. In practice, the feature map $\phi$ can be more complicated than the form $(2)$. For example, $\phi$ is a convolution layer in [1-3], and $\phi$ can be another pre-trained neural network in a line of practical works [4-6]. However, our analysis does not rely on the choice and complexity of $\phi$.
>
> >**Your comment**: 2. the dynamics seems to be simpler than DNN since all the layers share the same weights?
>
> **Our response**: Using weight-tying, the dynamics of an implicit neural network is governed by a fewer number of Gram matrices than explicit DNN, but it does not always mean it is simpler to analyze. The key challenge of implicit models is the well-posedness of forwarding propagation. It is easy to come up with a matrix $A$ that is not appropriately scaled, causing the non-existence of a fixed point $z^*$ even at initialization. As a result, the output and training may go to infinity too. Consequently, the gradient descent method cannot even start. Someone may be fortunate and choose a matrix $A$ to obtain a fixed point $z^*$ at initialization. However, Refs. [2,7,8] observed that the forward propagation takes increasingly more iterations and time to converge as the number of training epochs increases. Thus, it is possible that with an inappropriately scaled matrix $A$, its operator norm becomes increasingly larger. Consequently, the forward propagation diverges in some training epoch, and the fixed point $z^*(t)$ does not exist, causing the output and training loss to go to infinity, and the gradient descent method diverges. The main contribution of this paper is that by using a fixed scalar $\gamma$ at the beginning of the training, the well-posedness of forwarding propagation is guaranteed *during the entire training process*.
>
> [1] Trellis networks for sequence modeling, ICLR 2019.
>
> [2] Deep equilibrium models, NeurIPS 2019
>
> [3] Raft: Recurrent all-pairs field transforms for optical flow, ECCV 2020
>
> [4] Painting classification using a pre-trained convolutional neural network, ICCV 2016
>
> [5] Discriminating solitary cysts from soft tissue lesions in mammography using a pretrained deep convolutional neural network, Medical physics 2017
>
> [6] Voice conversion using RNN pre-trained by recurrent temporal restricted Boltzmann machines, TASLP 2014
>
> [7] Neural Ordinary Differential Equations, NeurIPS 2018
>
> [8] Stabilizing Equilibrium Models by Jacobian Regularization, ICML 2021

---

### Official Review · Reviewer_sNJZ · 2021-11-09

**Correctness:** 3
**Technical Novelty And Significance:** 2
**Empirical Novelty And Significance:** Not applicable
**Recommendation:** 3
**Confidence:** 4

**Main Review:**

The critical aspect of considering the convergence property for over-parameterized implicit networks is to show the non-singularity of the feature matrix $Z^*$, which is the fixed point of the non-linear equation $Z = \sigma(AZ+\phi(X))$, since we treat the final output as $Y= WZ^*$. This is a challenging and open problem for the community of theoretical implicit models. However, the submission considers a different output---$\hat{Y}= UZ^* + V\phi(X)$; hence there is no difficulty, and it is meaningless to get the smallest singular values as $\Theta(\lambda_0^{1/2})$, which is the same as previous over-parameterized explicit networks $\phi(X)$ and cannot show any difference between implicit and explicit DNNs. Unfortunately, the submission just got the results in this way.

1. The only difference between this submission and the previous works on explicit DNN convergence in the sense of proof roadmap is the additional proof for the existence of a fixed point at the initialization. However, constructing a shrink operator (which guarantees the existence of the fixed point) is not a complex task. We can even guarantee the well-posedness by setting $A(0)=0$. In fact, as the authors discussed in the submission, we need to prove that the operator $\sigma(A\cdot+\phi(x))$ is shrink during training rather than at initialization. For guaranteeing this, the scaling factor may depend on the other term, such as step size,  rather than only $m$, since we need to bound the difference $A(t)-A(0)$. The author needs to deal with the existence more carefully.
2. For the convergence speed, it is the same as the previous ones. Hence, it further verifies that the convergence guarantee comes from the explicit additional term $V\phi(X)$---two-layer over-param ReLU DNN, instead of the implicit feature $Z^*$. A straightforward guess is that all the results still hold when we set $A=0$, or set $U = 0$, or even drop $Z^*$, i.e., $\hat{Y}= V\phi(X)$.
3. When proving the non-singularity of $H$, the submission says that it utilizes a different data assumption---no two data are parallel to each other. However, the same setting and almost the same linear convergence results are given in [1].
4. More importantly, the current convergence guarantee for over-params DNNs can be divided into two categories in the sense of activation settings-- ReLU and sufficient smooth activation function. For proving the PL-inequality, one relies on the smoothness of activation to provide the lower bound, while the others prove the flipping feature is small and the overall bound can hold during training.  Confusingly, this submission mixes these two roadmaps, utilizes the routine for a smooth activation function in the ReLU setting, which may cause the problem for the conclusion of some auxiliary lemmas.

[1] Gradient Descent Provably Optimizes Over-parameterized Neural Networks.

**Summary Of The Paper:**

This submission considers a good problem but contributes a little.

**Summary Of The Review:**

consider a important problem, but heavily rely on the the results in the previous work.

---

> ### Author Response · Authors · 2021-11-19
> **Response to Reviewer sNJZ**
>
> Thank you for your constructive comments. We have carefully revised our paper according to your comments. Our point-to-point responses to your comments are as follows:
>
> >**Your concern**: 1. However, the submission considers a different output---$\hat{Y}=UZ^*+V\phi(X)$
> ; hence there is no difficulty, and it is meaningless to get the smallest singular values as $\Theta(\lambda_0^{1/2})$
> , which is the same as previous over-parameterized explicit networks and cannot show any difference between implicit and explicit DNNs.
>
> **Our response**: There are some misunderstandings on our model. The output we used in this paper (a combination of fixed point and feature vector) is commonly used in the community of implicit deep learning. For example, Eq. (1.1a) in [4] has the same output as ours and is a general form for the output of implicit models. Eqs. (5)-(6) in [9] have similar outputs as ours by using skip-connections to facilitate the training process, and Eq. (13) in [16] (with an identical form as ours) is one of the most commonly used formulations for learning a linear dynamic system.
> Moreover, more complicated mapping can be applied on the fixed point and feature vector at the output. For example, Eq. (7) in [10] applied another neural network on the hidden state for instance segmentation, Eq. (14) in [11] also used another neural network at the output for object detection, and Ref. [17] defines a quadratic energy function involving fixed point and output vectors for reasoning purposes.
>
> Also, there are fundamental differences between implicit and explicit neural networks. The key challenge in implicit neural networks is to maintain the well-posedness of the forward propagation *throughout the entire training*, which is *not* a common difficulty in explicit finite-depth neural networks. In particular, Ref. [12-15] all observe that, without appropriate scaling the weight matrices, the time spent on forwarding propagation gradually grows with training epochs. Thus, a line of works in [4, 14, 15] suggested taking extra efforts to resolve this problem, such as adding an extra softmax layer [15], solving a projection subproblem at each iteration [4], and adding Jacobian regularization [14]. In this work, instead of using these training-intrusive methods, we employ a fixed scalar $\gamma$ at the beginning of training to ensure the well-posedness overall training epochs. This scalar is initialized before training and kept fixed **throughout the entire training**. Thus, it can be seen that the proposed methods are different from previous works on explicit networks.
>
> >**Your concern**: 2. The only difference between this submission and the previous works on explicit DNN convergence in the sense of proof roadmap is the additional proof for the existence of a fixed point at the initialization. However, constructing a shrink operator (which guarantees the existence of the fixed point) is not a complex task. We can even guarantee the well-posedness by setting $A(0)=0$. In fact, as the authors discussed in the submission, we need to prove that the operator $\sigma(A\cdot+\phi(x))$ is shrink during training rather than at initialization. For guaranteeing this, the scaling factor may depend on the other term, such as step size, rather than only $m$, since we need to bound the difference $A(t)-A(0)$. The author needs to deal with the existence more carefully.
>
> **Our response**: Previous works usually employ some training-intrusive methods, such as extra softmax layer [15] for solving a projection subproblem in each iteration [1], or adding Jacobian regularization [14]. Our methods use a fixed scalar $\gamma$ that is initialized before training and kept fixed **throughout the entire training**. Thus, it is challenging to prove the convergence without changing $\gamma$ during the training. That is, the unique fixed point $z^*(t)$ exists not only at initialization but also exists in the entire training. In particular, we show for any matrix $A$ with a bounded operator norm, the unique fixed point exists if we use an appropriate scalar $\gamma$ to scale the matrix $A$ in Lemma 2.2. At initialization, it is easy to find such a scalar $\gamma$. However, the challenge is that the matrix $A(t)$ is updated and changed during the training process, and so is its operator norm $||A(t)||$. Thus, it is difficult to find a fixed scalar $\gamma$ at the beginning to scale all matrices $A(t)$ over all iterations. This is the reason that Ref. [4] reformulated the optimization problem and used a projection step in each iteration, Ref. [2] used an extra softmax layer, and Ref. [14] added a Jacobian regularization. However, in Section 3, we show the operator norms $A(t)$ is upper bounded by a constant $2c\sqrt{m}$ if $||A(0)||\leq c\sqrt{m}$ and the width $m$ is large. Therefore, by using $\gamma \leq \min(\gamma_0, \gamma_0/(2c))$ for any $0<\gamma_0<1$, the unique fixed point always exists throughout the entire training process.

---

> > ### Author Response · Authors · 2021-11-19
> > **Response to Reviewer sNJZ (continued)**
> >
> > (continued)
> >
> > >**Your concern**: 3. For the convergence speed, it is the same as the previous ones. Hence, it further verifies that the convergence guarantee comes from the explicit additional term $V\phi(X)$---two-layer over-param ReLU DNN, instead of the implicit feature $Z^*$. A straightforward guess is that all the results still hold when we set $A=0$, or set $U=0$, or even drop
> > , i.e., $\hat{Y}=V\phi(X)$.
> >
> > **Our response**: We believe there is some misunderstanding regarding the convergence results. The same convergence speed does not necessarily lead to the same results. The well-posedness of the forward propagation is the key challenge, even one uses skip connections or explicit additional terms. For example, at initialization, it is easy to come up with a matrix $A$ that is not appropriately scaled, so that the fixed point $z^*$ does not exist, e.g., $|z^{\ell}|\rightarrow \infty$ as $\ell\rightarrow \infty$. Since the output is the combination of the fixed point and feature vector, the output could also be infinitely large, i.e. $\hat{y}=\infty$, causing the training loss also to go to infinite. As a result, the gradient descent method cannot even start.
> > Although it is possible to obtain the fixed point $z^*$ at the initialization with a matrix $A$, the operator norm of $A$ could gradually grow with the training epochs, causing the forward propagation to take increasingly more iterations to converge, which are observed in [12-14]. As a result, the forward propagation may fail to converge, implying that a fixed point $z^*(t)$ does not exist, the output and training loss is infinitely large, and the gradient descent method diverges. Thus, the same convergence speed can lead to completely different results. Thus, your guess does not hold. But we still appreciate your insightful comments.
> >
> > >**Your concern**: 4. When proving the non-singularity of $H$, the submission says that it utilizes a different data assumption---no two data are parallel to each other. However, the same setting and almost the same linear convergence results are given in [1].
> >
> > **Our response**: Thanks for your comment. We would like to clarify that, although Ref. [1] uses the same assumption as ours, their settings are different from us. Specifically, Ref. [1] analyzes a two-layer neural network, while our paper focuses on an implicit neural network that could have infinitely many layers. Since Ref. [1] considers a two-layer sallow network, the well-posedness of the forward propagation is always guaranteed. However, the well-posedness for implicit models with infinite depth is the key challenge, which is what we are trying to address in our work. Thus, our work studies a completely different setting compared with Ref. [1].
> >
> > Moreover, the definition of matrix $H$ is different in these two papers and the proof techniques are also different in these two papers. In [1], they define $H:=(\Phi^{\prime})(\Phi^{\prime})^T$, while we define $H:=(\gamma^2M + I_n)\circ ZZ^T + Q(t)\circ XX^T+ \Phi\Phi^T$ in Eqns. (15-17). In addition, their proof relies on standard real and functional analysis to show that  $(\Phi^{\prime})(\Phi^{\prime})^T$ is strictly positive definite, while we use the Hermite expansions to show $(\Phi)(\Phi)^T$ is strictly positive definite so that $H$ is strictly positive definite. In fact, Ref. [7] also used the Hermite Expansions to show $(\Phi)(\Phi)^T$ is strictly positive definite but they assumed the data point $x$ follows some sub-Gaussian random vector distributions, while we only assume no two data points are parallel to each other.

---

> > > ### Author Response · Authors · 2021-11-19
> > > **Response to Reviewer sNJZ (continued)**
> > >
> > > >**Your concern**: 5. More importantly, the current convergence guarantee for over-params DNNs can be divided into two categories in the sense of activation settings-- ReLU and sufficient smooth activation function. For proving the PL-inequality, one relies on the smoothness of activation to provide the lower bound, while the others prove the flipping feature is small and the overall bound can hold during training. Confusingly, this submission mixes these two roadmaps, utilizes the routine for a smooth activation function in the ReLU setting, which may cause the problem for the conclusion of some auxiliary lemmas.
> > >
> > > **Our response**: Since $\sigma^{\prime}$ is not Lipschitz continuous, the first category does not apply in our case. In general, if the derivative of the activation is not Lipschitz continuous, the proofs become much more complicated and involved. In particular, one has to study quantities related to the changes of the activation patterns during training such as [3, 5]. However, [6] claims this analysis is not necessary by introducing an alternative proof. Specifically, [6] suggests training all layers of the network, and focusing the analysis on the last layer. As a result, they are able to establish the linear global convergence result for finite-depth neural networks if the network is overparameterized and there is no need to bound the number of sign flips of the activation neurons, nor the Lipschitz constant of the gradient. Unfortunately, their study cannot directly apply to an implicit neural network since an implicit neural network could have infinite may layers, and the well-posedness of forwarding propagation could be a problem. In this paper, we adapt their strategy to an implicit model (with infinite many layers) and show the well-posedness can be resolved by using a fixed scalar $\gamma$ at beginning of the training. As a result, the global convergence for the gradient descent for an implicit model is established.
> > >
> > > [1] Gradient Descent Provably Optimizes Over-parameterized Neural Networks, ICLR 2018
> > >
> > > [2] On the Theory of Implicit Deep Learning: Global Convergence with Implicit Layers, ICLR 2021
> > >
> > > [3] A Convergence Theory for Deep Learning via Over-Parameterization, ICML 2019
> > >
> > > [4] Implicit Deep Learning, SIMODS 2021
> > >
> > > [5] An Improved Analysis of Training Over-parameterized Deep Neural Networks, NeurIPS 2019
> > >
> > > [6] On the Proof of Global Convergence of Gradient Descent for Deep ReLU Networks with Linear Widths, ICML 2021
> > >
> > > [7] Global Convergence of Deep Networks with One Wide Layer Followed by Pyramidal Topology, NeurIPS 2020
> > >
> > > [8] Gradient descent finds global minima of deep neural networks, ICML 2019
> > >
> > > [9] Robust Learning with Implicit Residual Networks, MAKE 2021
> > >
> > > [10] Implicit Feature Refinement for Instance Segmentation, MM 2021
> > >
> > > [11] Implicit Feature Pyramid Network for Object Detection, arXiv 2020
> > >
> > > [12] Neural Ordinary Differential Equations, NeurIPS 2018
> > >
> > > [13] Deep equilibrium models, NeurIPS 2019
> > >
> > > [14] Stabilizing Equilibrium Models by Jacobian Regularization, ICML 2021
> > >
> > > [15] On the Theory of Implicit Deep Learning: Global Convergence with Implicit Layers, ICLR 2021
> > >
> > > [16] Contraction-Based Methods for Stable Identification and Robust Machine Learning: a Tutorial, arXiv 2021
> > >
> > > [17] Understanding Deep Architectures with Reasoning Layer, NeurIPS 2020
> > >
> > > [18] Toward Moderate Overparameterization: Global Convergence Guarantees for Training Shallow Neural Networks, IEEE JSAIT 2019.
> > >
> > > [19] Tight Bounds on the Smallest Eigenvalue of the Neural Tangent Kernel for
> > > Deep ReLU Networks, ICML 2021.
> > >
> > > [20] Neural tangent kernel: Convergence and generalization in neural networks, NeurIPS 2018.
> > >
> > > [21] On the convergence rate of training recurrent neural networks, NeurIPS 2019.
> > >
> > > [22] Stochastic Gradient Descent Optimizes Over-parameterized Deep ReLU Networks, Machine Learning 2019.
> > >
> > > [23] Deep Residual Learning for Image Recognition, CVPR 2016.
> > >
> > > [24] Width Provably Matters in Optimization for Deep Linear Neural Networks, ICML 2019
> > >
> > > [25] Global convergence of gradient descent for deep linear residual networks, NeurIPS 2019.
> > >
> > > [26] Convergence of adversarial training in overparametrized neural networks, NeurIPS 2019.
> > >
> > > [27] Fast Convergence of Natural Gradient Descent for Overparameterized Neural Networks, NeurIPS 2019.
> > >
> > > [28] Over-parameterized Adversarial Training: An Analysis Overcoming the Curse of Dimensionality, NeurIPS 2020.
> > >
> > > [29] On the Global Convergence of Training Deep Linear ResNets, ICLR 2020.
> > >
> > > [30] Optimization of Graph Neural Networks: Implicit Acceleration by Skip Connections and More Depth, ICML 2021

---

> > > ### Comment · Reviewer_sNJZ · 2021-11-30
> > > **About the proof the non-singularity of Gramm matrix**
> > >
> > > Although the authors used a different technique with [1], the submission only presented the same result as in [1] without any new insight. Moreover, this technique is the same as in [7]. Indeed, [7] assumes sub-Gaussian data, but it is because that, [7] aims to figure out the exact quantity of \lambda. Without the assumption of sub-Gaussian data, the non-singularity of the Gramm matrix still can be proved using their framework.  Therefore, I believe the contribution of this submission is incremental.

---

> > > > ### Author Response · Authors · 2021-12-02
> > > > **Response to Reviewer sNJZ about the proof the non-singularity of Gramm matrix**
> > > >
> > > > >**Your comment**: `Although the authors used a different technique with [1], the submission only presented the same result as in [1] without any new insight. Moreover, this technique is the same as in [7]. Indeed, [7] assumes sub-Gaussian data, but it is because that, [7] aims to figure out the exact quantity of $\lambda$. Without the assumption of sub-Gaussian data, the non-singularity of the Gramm matrix still can be proved using their framework. Therefore, I believe the contribution of this submission is incremental.
> > > >
> > > > **Our response**: We cannot agree with the reviewer's comment ``without any new insight". In fact, many works [2,3,5,6,7,8] achieved the same convergence result as [1]. If the reviewer's point were correct, [2,3,5,6,7,8] are not making any contribution to the community. Just similar to our work, [2,3,5,6,7,8] make significant contributions to the community because we are working in totally different settings. In particular, the result in [1] holds only for a \textbf{two-layer} neural network, the results in [3,5,6,7,8] hold for a **finite-depth** neural network, the result in [2] holds for a **linear** implicit neural network. In our work, we established the results for a **nonlinear implicit neural network**.
> > > >
> > > > To make it clear, we list detailed differences between [1] and our paper. Firstly, [1] works on a two-layer
> > > > neural network, while we are working on an implicit neural network that could have infinitely many layers; Although we make
> > > > the same assumption on data matrix $X$, [1] proves $(\Phi)^{\prime}(\Phi^{\prime})^T$ is strictly positive definite w.h.p, while we proves $\Phi\Phi^T$ is strictly positive definite w.h.p; Since [1] works on two-layer neural network, [1] does not need to worry about \textbf{well-posedness issue}, which poses significant challenges to the training process in implicit neural networks,
> > > > while we solve the well-posedness throughout the entire training by using a fixed scalar at the beginning of training.
> > > >
> > > > Secondly, [7] does not provide the exact quantity of $\lambda$ (or $\lambda_*$ defined in equation (13) of [7]) under the
> > > > assumption of sub-Gaussian. Instead, [Theorem~3.3, 7] shows that $\lambda_*>b_1$ with high probability, but $b_1>0$ is some
> > > > constant which is independent of sample size $N$ and input dimension $d$. However, $b_1$ depends on integer $k$ under the assumption of $N\leq d^k$. Thus, in a general case, $b_1$ depends on $N$, so we don't know the exact quality of $\lambda_*$.
> > > > Instead, we only know such $b_1>0$ exists and $\lambda_*>0$ w.h.p, which is the basic result and pursuit in the previous works along this line [1,3,5,6,7,18,19]. Moreover, the techniques used in [7] are not new. Instead, [7] follows the framework proposed in [18] but with different setups and assumptions. In particular, [18] works on two-layer neural network assuming $\min\{||x_i-x_j||,||x_i+x_j||\}\geq \delta$ for all $ i,j\in [N]$, [7] works on finite-depth neural networks assuming the sub-Gaussian distribution of $x_i$, while we are working on an implicit neural network that could have infinite-many layers and assume no two data are parallel to each other. We all made some assumptions on the data so as to show the non-singularity of the Gram matrix. Thus, we cannot agree with the reviewer's comment "Without the assumption of sub-Gaussian data, the non-singularity of the Gram matrix still can be proved". Instead, data assumptions are commonly used in order to show the non-singularity of the Gram matrix among different frameworks. For example, [1,8] and us assume non-parallel data points, [6,7,19] assume the sub-Gaussian distribution of $x_i$, [18] assumes $\min \{||x_i-x_j||, ||x_i+x_j|| \}\geq \delta$, [3,5] relaxes the assumption made by [18] to assume $||x_i-x_j||\geq \delta$, and [19] assumes the distribution of $x_i$ satisfies Lipschitz concentration for any Lipschitz continuous function. Thus, we believe such data assumption is necessary and the techniques used in [7] and this paper are not exactly the same.

---

> > ### Comment · Reviewer_sNJZ · 2021-11-30
> > **The main concern is the theoretical contribution**
> >
> > Really thanks for the careful reply!!
> >
> > 1. My main concern is that the formulation makes the theoretical contribution incremental, NOT whether it is common. The additional explicit term makes the proof and analysis the same as previous over-parameterized explicit networks. I point out some similarities, such as the convergence rate and the details in the proof. The author should compare these places, for example, why the convergence rate is the same as explicit DNN's, what will happen if we set U=0.
> >
> > 2. The theoretical contribution on shrink operator, repeatedly emphasized here, is only an additional product of the previous conclusion: (1) with proper scaling, we can easily get a shrink operator at the init point; (2) then when the network is wide enough, the training iterate is quite close to the init points (this is proven in many previous papers); (3) hence, use the scale constant with the same order of at the initialization we can bound all the iterates. From this point, using a gaussian matrix concentration property at step (1) cannot be considered a significant contribution.

---

> > > ### Author Response · Authors · 2021-12-02
> > > **Response to Reviewer sNJZ about the theoretical contribution**
> > >
> > > >**Your comment**: My main concern is that the formulation makes the theoretical contribution incremental, NOT whether it is common. The additional explicit term makes the proof and analysis the same as previous over-parameterized explicit networks. I point out some similarities, such as the convergence rate and the details in the proof. The author should compare these places, for example, why the convergence rate is the same as explicit DNN's, what will happen if we set U=0.
> > >
> > > **Our comment**: We believe the reviewer underestimates the contributions of our work. The formulation adopted in our work is commonly used and does not fundamentally change the settings we are working with: **infinite-depth non-linear implicit neural network**. All previous works obtained the results either for finite-depth neural networks or linear implicit neural networks. Our work is based on a completely different setting and previous techniques cannot be directly applied due to infinite depth and non-linearity. It is not fair to say a work's contribution is incremental just because it uses some previous techniques. As researchers, we are all standing on the shoulders of giants.
> > >
> > > For the similarity of convergence results, many works [2,3,5,6,7,8] achieved the same convergence result as [1]. In particular, the result in [1] holds only for a **two-layer** neural network, the results in [3,5,6,7,8] hold for a **finite-depth** neural network, the result in [2] holds for
> > > a **linear** implicit neural network. In our work, we established the results for a **nonlinear implicit neural network**. For the similarity of details in the proof, all the previous works [1,2,3,5,6,7,8,18,19] follow almost the same steps to establish the global convergence, since [20] points out training an overparameterized neural network can be expressed as a kernel gradient. Thus, the standard strategy is to show the Gram matrix is non-singular at initialization w.h.p and the Gram matrix at the latter iterations is close to its initialization so that the spectral property of the Gram matrix is preserved w.h.p. Although they use the almost same strategy, their settings are different. Specifically, [1] consider a two-layer ReLU activated neural network trained by GD, [18] also consider two-layer neural network trained by GD but using smooth activation function, [8] focuses on a finite-depth neural network with smooth activation function trained by GD, [7] obtains the same result as in [8] but only use one wide layer followed by a pyramidal topology, [3,5] focuses on a finite-depth ReLU activated neural network trained by (S)GD, [6] establish the same result as in [3,5] but only use linear widths, and [2] focuses on linear implicit neural network trained by gradient flow. With different setups in terms of the optimizer, neural network architecture, and better bounds, this standard strategy is applied in more recent works [21,22,24-30]. Similar to our work, [2,3,5,6,7,8,21,22,24-30] also make
> > > significant contributions to the community because we are working in totally different settings compared to [1]. We hope the reviewer can notice these important differences instead of the convergence results and overall strategy.
> > >
> > > >**Your comment**: ``The theoretical contribution on shrink operator, repeatedly emphasized here, is only an additional product of the previous conclusion: (1) with proper scaling, we can easily get a shrink operator at the init point;  (2) then when the network is wide enough, the training iterate is quite close to the init points (this is proven in many previous papers); (3) hence, use the scale constant with the same order of at the initialization, we can bound all the iterates. From this point, using a gaussian matrix concentration property at step (1) cannot be considered a significant contribution."
> > >
> > > **Our response**: We thank the reviewer for concisely summarizing our proof. Apparently, we solve the well-posed problem in implicit neural networks in a simple yet elegant way. Previous methods employ projected gradient descent [4], softmax layer [2], and Jacobian regularization [8] to solve the well-posed problem. Our method just uses a fixed scalar to solve this problem, which provides great simplification for both practical usage and mathematical analysis perspectives. We believe such kind of simple yet elegant methods such as skip connection in ResNet [23] are making fundamental contributions to the community.

---

### Decision · Program_Chairs · 2022-01-20

**Decision:**

Accept (Poster)

**Comment:**

This paper shows gradient flow of ReLU activated implicit networks converges to a global minimum at a linear rate for the square loss when the implicit neural network is over-parameterized. While the analyses follow the existing NTK-type analyses and there are disagreements among reviewers on the novelty of this paper, the meta reviewer values new theoretical results on new, emerging settings (implicit neural networks), and thus decides to recommend acceptance